# SARS-CoV-2 sensing by RIG-I and MDA5 links epithelial infection to macrophage inflammation

Lucy G Thorne[†],[*] , Ann-Kathrin Reuschl[†] , Lorena Zuliani-Alvarez[†] , Matthew V X Whelan[†] , Jane Turner , Mahdad Noursadeghi , Clare Jolly[‡],[**] & Greg J Towers[‡],[***]

## Abstract

**SARS-CoV-2 infection causes broad-spectrum immunopathological disease, exacerbated by inflammatory co-morbidities. A better understanding of mechanisms underpinning virus-associated inflammation is required to develop effective therapeutics. Here, we discover that SARS-CoV-2 replicates rapidly in lung epithelial cells despite triggering a robust innate immune response through the activation of cytoplasmic RNA sensors RIG-I and MDA5. The inflammatory mediators produced during epithelial cell infection can stimulate primary human macrophages to enhance cytokine production and drive cellular activation. Critically, this can be limited by abrogating RNA sensing or by inhibiting downstream signalling pathways. SARS-CoV-2 further exacerbates the local inflammatory environment when macrophages or epithelial cells are primed with exogenous inflammatory stimuli. We propose that RNA sensing of SARS-CoV-2 in lung epithelium is a key driver of inflammation, the extent of which is influenced by the inflammatory state of the local environment, and that specific inhibition of innate immune pathways may beneficially mitigate inflammation-associated COVID-19.**

**Keywords** epithelial; inflammation; macrophage; RNA sensing; SARS-CoV-2
**Subject Categories** Immunology; Microbiology, Virology & Host Pathogen Interaction
**The EMBO Journal (2021) 40: e107826**

## Introduction

SARS-CoV-2 has caused a devastating pandemic, > 74.8 million confirmed cases, > 1.6 million deaths (https://covid19.who.int/, 20 December 2020) and a worldwide economic crisis. Infection causes a remarkably wide, but poorly understood, disease spectrum, ranging from asymptomatic (Allen *et al*, 2020; Treibel *et al*, 2020) to severe acute respiratory distress syndrome, multi-organ failure and death (Docherty *et al*, 2020; Zhou *et al*, 2020).

The success of immunosuppressive corticosteroid dexamethasone in treating COVID-19 (Beigel *et al*, 2020) suggests the importance of immunopathology in disease, likely driven by immune activation in infected and virus-exposed cells. Intracellular innate immune responses have evolved to detect and suppress invading pathogens, but inappropriate responses can also contribute to disease (Blanco-Melo *et al*, 2020; Park & Iwasaki, 2020). Pathogen-associated molecular patterns (PAMPs) are detected by pattern recognition receptors (PRR), including cytoplasmic nucleic acid sensors, and Toll-like receptors (TLR) that sample extracellular and endosomal space. PRR activation triggers signalling cascades which activate downstream transcription factors, including interferon (IFN) regulatory factors (IRFs) and NF-κB family members, to initiate a defensive pro-inflammatory gene expression programme, principally mediated by IFN secretion from infected cells. Paracrine and autocrine IFN signalling can suppress viral replication and spread and, together with other secreted cytokines and chemokines, coordinates adaptive immune responses. Viruses have evolved countermeasures to innate defences and deploy a combination of evasion, and direct innate immune pathway antagonism, to promote replication (Sumner *et al*, 2017). The resulting virus–host conflict is often a significant cause of pathogenesis with PRR-induced inflammation driving disease at the site of replication and systemically (Park & Iwasaki, 2020).

Missense mutations in innate immune pathways (Pairo-Castineira *et al*, 2020; Zhang *et al*, 2020), and autoantibodies leading to deficient type 1 IFN responses (Bastard *et al*, 2020), are associated with severe COVID-19, suggesting that intact innate immune responses are important in preventing disease, probably through controlling viral replication. Co-morbidities linked to severe disease are typically inflammatory in nature, suggesting that certain types of pre-existing inflammation influence disease severity (Paranjpe *et al*, 2020). However, the specific host–pathogen interactions that cause disease, and how these are impacted by existing inflammation, are not understood. Identification of the molecular events that link viral replication to inflammation and disease will be critical in the development of novel and more precise therapeutic agents. Moreover, such new knowledge will provide insights into

Division of Infection and Immunity, University College London, London, UK
*Corresponding author. Tel: +44 0203 108 2422; E-mail: l.thorne@ucl.ac.uk
**Corresponding author. Tel: +44 0203 108 2138; E-mail: c.jolly@ucl.ac.uk
***Corresponding author (lead contact). Tel: +44 0203 108 2112; E-mail: g.towers@ucl.ac.uk
[†]These authors contributed equally to this work
[‡]These authors contributed equally to this work as senior authors

the mechanisms by which the associated risk factors for severe COVID-19 impact immune homeostasis in general.

Here, we investigated early host–virus interactions to understand the mechanisms by which SARS-CoV-2 induces an innate response, whether it can escape consequent innate immune control and how it may propagate an immunopathogenic response. We focussed on lung epithelial cells and primary macrophages, which represent cells responsible for the earliest innate immune response to the virus (Bost *et al*, 2020; Chua *et al*, 2020). We found rapid replication and infectious virus release in lung epithelial cells prior to potent innate immune activation. Indeed, the cocktail of soluble mediators produced by infected cells strongly activated macrophages, which propagated a pro-inflammatory response. Critically, the production of an inflammatory secretome was directly downstream of RNA sensing by RIG-I and MDA5 because manipulation of sensing or signalling events in infected cells, using RNA interference or signalling pathway inhibition, suppressed subsequent macrophage activation and inflammatory gene expression. Furthermore, pre-exposure of epithelial cells or macrophages to exogenous inflammatory stimuli exacerbated inflammatory responses upon SARS-CoV-2 exposure. We propose that the innate immune microenvironment, in which sensing of SARS-CoV-2 infection occurs, determines the degree of virus-induced inflammation and has the potential to drive disease.

# Results

## SARS-CoV-2 activates delayed innate immune responses in lung epithelial cells

In order to investigate innate immune responses to SARS-CoV-2, we first sought a producer cell line that did not respond to the virus, thereby allowing production of virus stocks free of inflammatory cytokines. As adaptive mutations have been reported during passage of the virus in Vero.E6 cells (Davidson *et al*, 2020; Ogando *et al*, 2020), we selected human gastrointestinal Caco-2 cells, which express the SARS-CoV-2 receptor ACE2 and entry factors TMPRSS2/4 (Fig EV1A and B) and are naturally permissive (Stanifer *et al*, 2020). We found that Caco-2 support high levels of viral production (Fig EV1C and D), but not virus spread (< 15% cells infected) (Fig EV1E and F). Importantly, they do not mount a detectable innate response to SARS-CoV-2 over 72 hpi at a range of multiplicities of infection (MOIs), as evidenced by a lack of interferon-stimulated gene induction (ISG) (Fig EV1G). They are also broadly less responsive to innate immune agonists than lung epithelial Calu-3 cells (compare Fig EV1H-Caco-2 and Fig EV1I-Calu-3). Caco-2 cells were therefore used to produce SARS-CoV-2 stocks uncontaminated by inflammatory cytokines.

Comparatively, lung epithelial Calu-3 cells express high levels of receptor ACE2, and entry co-factors TMPRSS2 and TMPRSS4 (Fig EV1A and B) (Hoffmann *et al*, 2020; Zang *et al*, 2020), and are innate immune competent (Fig EV1I) when stimulated with various PRR agonists. Consistently, Calu-3 cells supported very rapid spreading infection of SARS-CoV-2 followed by the activation of innate immune responses. SARS-CoV-2 replication displayed > 1,000-fold increase in viral genomic and subgenomic (envelope, E) RNA levels within 5 h post-infection (hpi) across a range of MOIs 0.08, 0.4, 2 TCID50/cell (Figs 1A and EV2A), with TCID50

determined in Vero.E6 cells. Genomic and subgenomic E RNA in Calu-3 plateaued around 10 hpi. Rapid spreading infection was evidenced by increasing nucleocapsid protein (N)-positive cells by flow cytometry and immunofluorescence staining, peaking at 24 hpi with 50-60% infected cells (Figs 1B–D and EV2B). Infectious virus was evident in supernatants by 5 hpi at the highest MOI and peaked between 10–48 hpi, depending on MOI (Figs 1E and EV2C). A pronounced innate immune response to infection followed the peak of viral replication, evidenced by induction of cytokines (IL-6, TNF), chemokines (CCL2, CCL5) and type I and III IFNs (IFNβ, IFNλ1/3) measured by RT–qPCR (Figs 1F and G, and EV2D–F). This was accompanied by an IFN-stimulated gene (ISG) expression signature (CXCL10, IFIT1, IFIT2, MxA) (Figs 1H and EV2D–F). Gene induction was virus dose-dependent at 24 hpi, but equalised across all MOIs by 48 hpi, as the antiviral response to low-dose virus input maximised. These data show that infected lung epithelial cells can be a direct source of inflammatory mediators.

We were surprised that SARS-CoV-2 replicated so efficiently in Calu-3 despite innate immune responses including IFN and ISG expression because coronaviruses, including SARS-CoV-2, are reported to be IFN sensitive (Stanifer *et al*, 2020). Indeed, recombinant type I IFN, but not type II or type III IFNs, effectively reduced SARS-CoV-2 replication if Calu-3 cells were treated prior to infection (Figs 1I–K and EV2G and H). However, type I IFN had little effect on viral replication when added 2 h after infection (Fig 1I–K). Thus, the IFN response induced in infected lung epithelial Calu-3 cells appears too late to suppress SARS-CoV-2 replication in this system. To determine whether viral exposure dose influences the race between viral replication and IFN, we infected cells at a 100× lower dose (MOI 0.0004 TCID50/cell) and observed a longer window of opportunity for exogenous type I IFN to restrict viral replication (Fig 1I–K). This is consistent with the hypothesis that high-dose infection can overcome IFN-inducible restriction.

## Peak SARS-CoV-2 replication precedes innate immune activation

To understand the apparent disconnect in the kinetics between innate immune activation and viral replication, we used single-cell imaging to measure nuclear localisation of activated inflammatory transcription factors NF-κB p65 and IRF3, which mediate multiple PRR-signalling cascades. NF-κB p65 nuclear translocation coincided with cells becoming N protein positive, and a change was evident from 5 hpi (Figs 2A and B, and EV3). The timing of NF-κB p65 translocation was dependant on the viral dose, from 5 hpi for the highest MOI (2 TCID50/cell, Fig EV3), between 5 and 10 hpi for MOIs 0.4 and 0.04 (Figs 2A and B, and EV3), and 24–48 hpi for MOI 0.004 (Fig EV3). IRF3 activation was also virus dose-dependent but did not maximise until 72 hpi, later than NF-κB (Fig 2C and D), and we observed a more modest shift in IRF3 nuclear intensity compared with NF-κB throughout infection. These data are consistent with the requirement of a threshold of viral RNA replication to induce transcription factor translocation and innate immune activation and suggest that SARS-CoV-2 may antagonise IRF3 activation to a greater extent than NF-κB. Although small variation in NF-κB p65 and IRF3 nuclear intensity was observed in N-negative cells, we did not see the same large increases sustained throughout the time course as in N-positive cells, consistent with direct activation of NF-κB p65 and IRF3 by virus replication (Fig EV3).

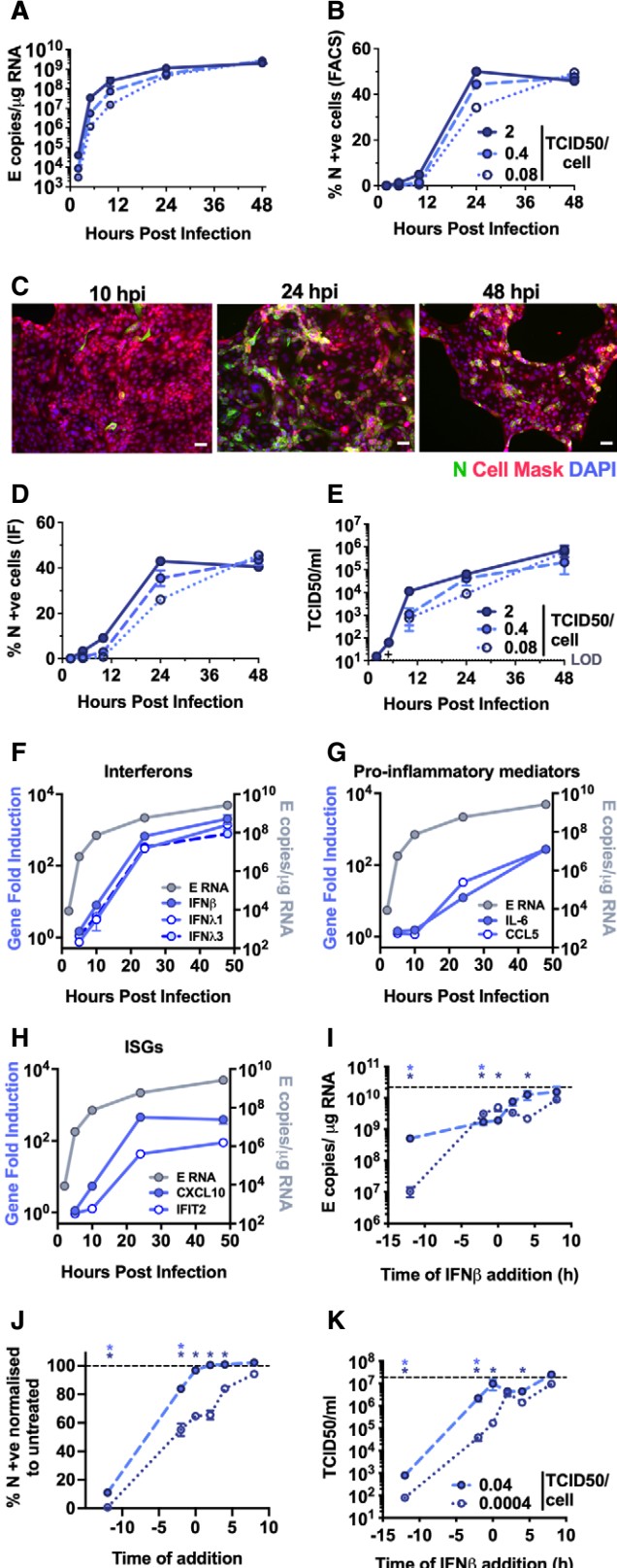

**Figure 1. SARS-CoV-2 activates delayed innate immune responses in lung epithelial cells.**

A–H Measurements of replication and innate immune induction in Calu-3 lung epithelial cells infected with SARS-CoV-2 at MOIs 0.08, 0.4 and 2 TCID50$_{VERO}$/cell. (A) Replication of SARS-CoV-2 genomic and subgenomic E RNAs (qRT–PCR). (B) Quantification of N staining from cells in (A) by flow cytometry. Mean percentage of N positive of all live-gated cells is shown ± SEM, n = 2. (C) Representative example of immunofluorescence staining of N protein (green) after SARS-CoV-2 infection of Calu-3 at MOI 0.4 TCID50$_{VERO}$/cell, at time points shown. Nuclei (DAPI, blue), cell mask (red). Scale bar represents 50 μm. (D) Quantification of N staining in cells in (C) by immunofluorescence. (E) Infectious virus released from cells in (A) determined by TCID50 on Vero.E6 cells. (F-H) Fold induction of (F) interferons (IFNβ, IFNλ1 and IFNλ3) (G) pro-inflammatory mediators (IL-6 and CCL5) or (H) IFN-stimulated genes (CXCL10 and IFIT2) each overlaid with SARS-CoV-2 E (qRT–PCR). All data from cells in (A) at MOI 0.4 TCID50$_{VERO}$/cell. (A–H) Means from replicate wells shown ± SEM n = 2; full growth curve is representative of three independent experiments.

I–K SARS-CoV-2 infection (MOIs 0.04 (closed symbols) and 0.0004 (open symbols) TCID50$_{VERO}$/cell) in Calu-3 cells with addition of 10 ng/ml IFNβ before or after infection at time points shown, measured by (I) E RNA copies (J) N-positive cells, (K) released virus (TCID50$_{VERO}$/cell) all measured at 24 hpi. Dotted line indicates untreated. Mean ± SEM, n = 3, one-way ANOVA light and dark blue * indicates significance for high and low MOIs, respectively.

Supporting the observation of activation of NF-κB p65 and IRF3 activation by SARS-CoV-2 infection, single-cell fluorescence *in situ* hybridisation (FISH) analysis of IL-6 mRNA (a prototypic NF-κB regulated cytokine) showed increased IL-6 transcripts uniquely in N-positive infected cells, appearing at 6 hpi and peaking at 24 hpi (Figs 2E and F, and EV4A). IFIT1 transcripts (a prototypic ISG) measured by FISH also demonstrated rapid induction in N-positive cells with increased signal from at 6 hpi (Fig 2G and H). Strikingly, IFIT1 mRNA was not highly induced in N-negative bystander cells consistent with defective interferon responses failing to induce ISGs and a timely antiviral state in uninfected cells (Fig 2H and I). As a control for these changes, we show that GAPDH transcripts did not change (Fig EV4B). Secretion of pro-inflammatory chemokine CXCL10, and cytokine IL-6, followed gene expression and was detected from 24 hpi (Figs 2J and K, and EV4C). Further analysis revealed increases in lactate dehydrogenase (LDH) in infected cell supernatants from 48 hpi, equal across all MOIs, indicative of pro-inflammatory cell death (Figs 2L and EV4D). Importantly, cytokine secretion had also equalised across MOIs from 24 hpi (Fig 2J and K). LDH release paralleled loss of the epithelial monolayer integrity (Fig 1C) and cell death (Figs 2M and EV4E and F) accounting for the reduction in cytokine secretion at 72 hpi (Fig 2J and K).

**SARS-CoV-2 is sensed by MDA5 and RIG-I**

To determine the mechanism of virus sensing by innate pathways, we first confirmed that viral RNA replication is required for innate immune activation. Inhibition of viral RNA replication, with polymerase inhibitor Remdesivir, abrogated pro-inflammatory and ISG gene expression in a dose-dependent manner (Fig 3A–D). Critically,

**Figure 1.**

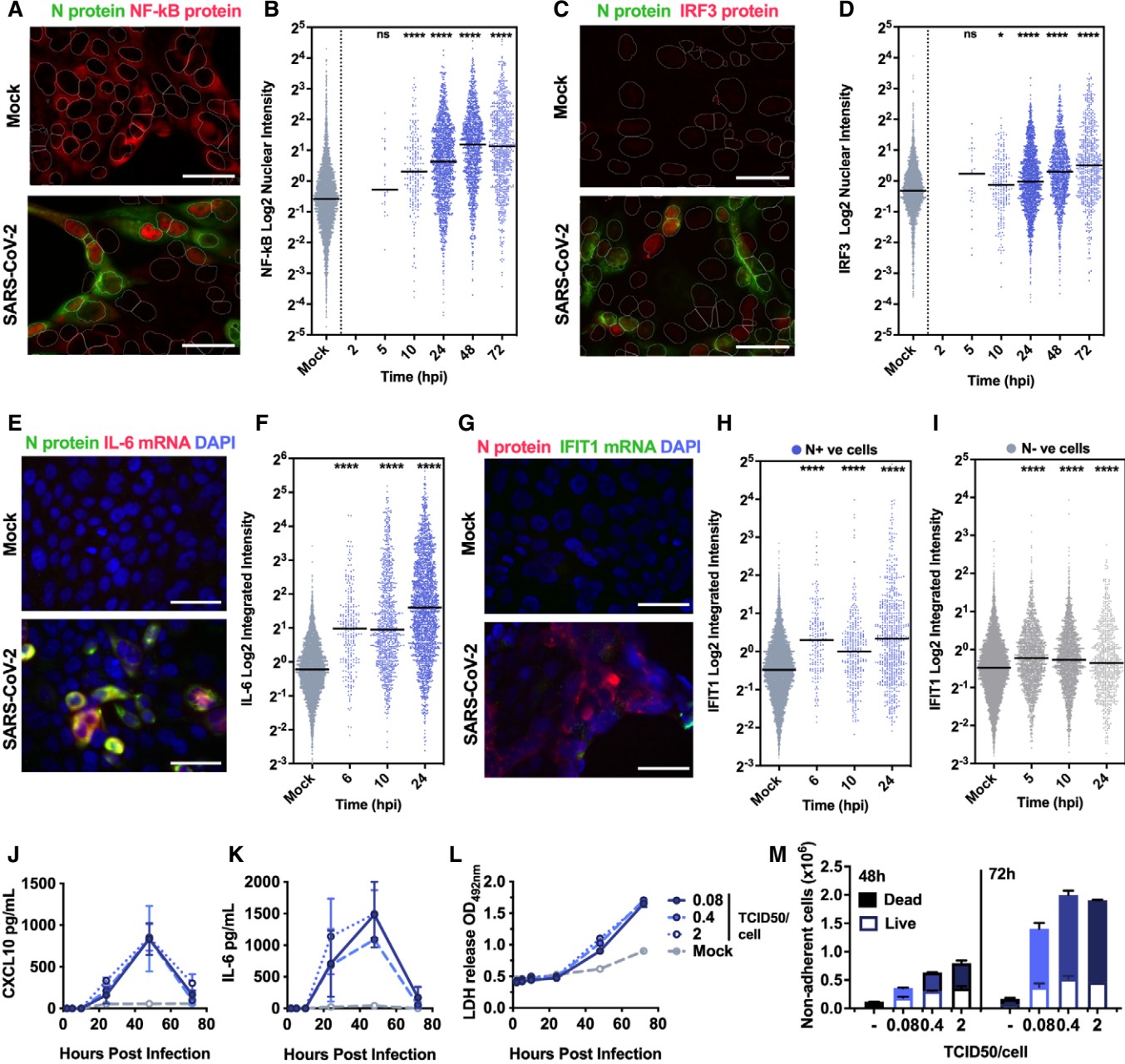

**Figure 2. Peak SARS-CoV-2 replication precedes innate immune activation.**

A–I  (A, C) Representative images of NF-κB p65 (A) (red) and IRF3 (C) (red) nuclear localisation in mock or SARS-CoV-2-infected (MOI 0.4 TCID50_VERO/cell) Calu-3 cells at 24 hpi. SARS-CoV-2 N protein (green). (E and G) Representative images of IL-6 mRNA (E) detected by FISH (red) and N protein (green), or IFIT1 mRNA (G) (green) with N protein (red), both with nuclei (DAPI, blue) in mock or SARS-CoV-2-infected (MOI 0.4 TCID50_VERO/cell) Calu-3 cells at 24 hpi. (B, D, F, H, I) Single-cell analysis time course quantifying the integrated nuclear intensity of NF-κB p65 (B), IRF3 (D), or overall integrated intensity for IL-6 (F) or IFIT1 (H) mRNA over time in N protein-positive cells and N protein-negative cells (I). $n = 2$. Kruskal–Wallis test with Dunn's multiple comparison. * ($P < 0.05$), **** ($P < 0.0001$). Scale bar represents 50 μm.

J, K  Secretion of CXCL10 (J) and IL-6 (K) by infected Calu-3 cells (MOIs 0.08, 0.4 and 2 TCID50_VERO/cell), (ELISA).

L  Lactate dehydrogenase (LDH) release into culture supernatants by mock and SARS-CoV-2-infected Calu-3 cells (MOIs 0.08, 0.4 and 2 TCID_VERO50/cell) quantified absorbance (492nm).

M  Quantification of live/dead staining of non-adherent cells recovered from supernatants of mock or SARS-CoV-2-infected Calu-3 cultures at 48 and 72 hpi.

Data information: (J–M) Means from replicate wells shown ± SEM, $n = 2$, representative of three independent experiments.

Remdesivir was only effective if added prior to, or at the time of infection, consistent with a requirement for metabolism to its active tri-phosphorylated form (Eastman *et al*, 2020) (Fig 3E–H).

Inflammatory gene induction dependent on viral genome replication suggested that an RNA sensor activates this innate response.

Both genomic and subgenomic SARS-CoV-2 RNAs are replicated via double-stranded intermediates in the cytoplasm (Li *et al*, 2021). Accordingly, we detected cytoplasmic dsRNA at 5 hpi in Calu-3 cells, preceding N positivity (Fig 3I) and by 48 hpi all dsRNA-positive cells were N positive. Depletion of RNA sensing adaptor

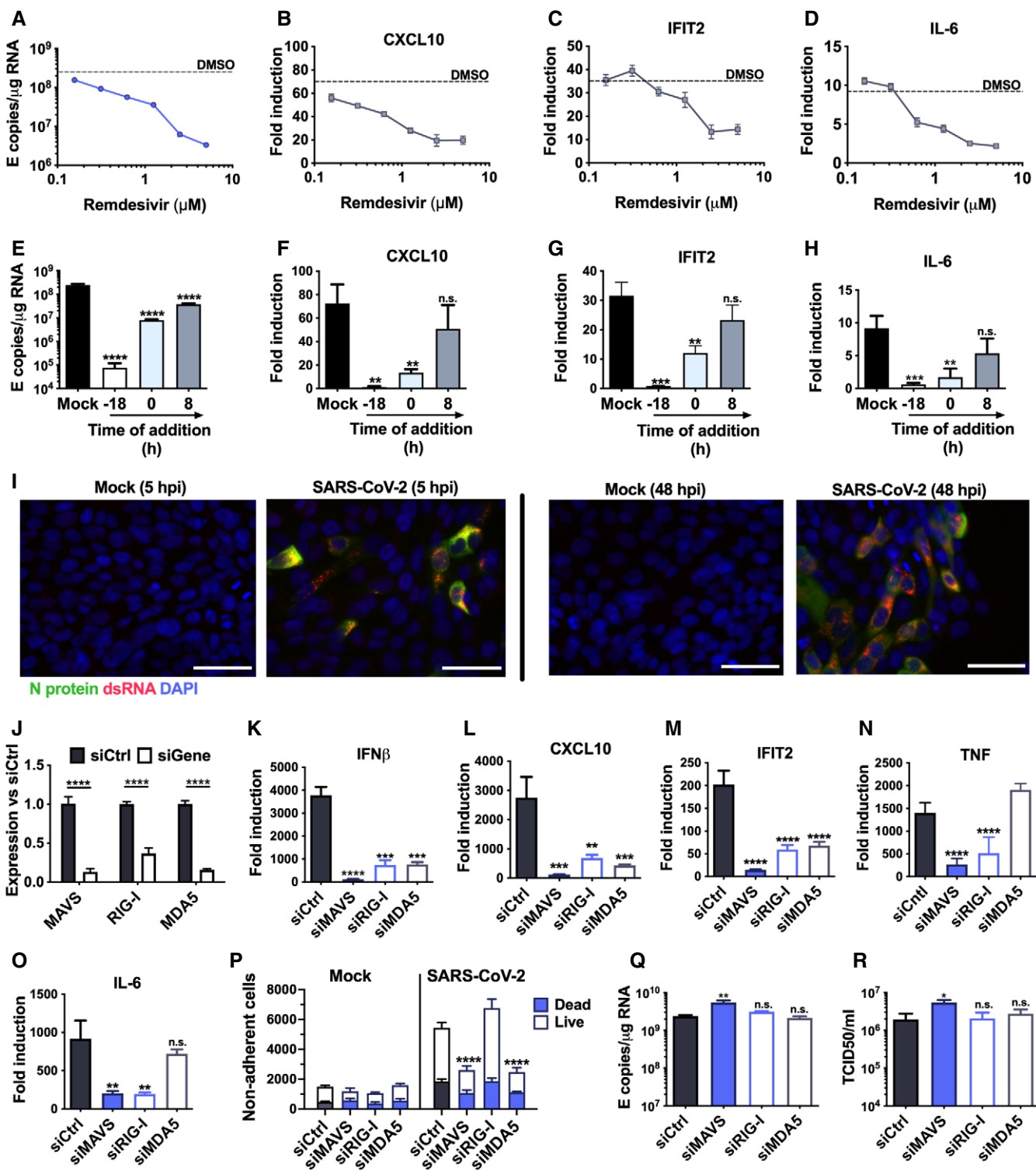

**Figure 3.**

**Figure 3. SARS-CoV-2 is sensed by MDA5 and RIG-I.**

A–D   Measurement of (A) viral genomic and subgenomic E RNA at 24 hpi, (B) fold induction of CXCL10 from (A), (C) IFIT2 and (D) IL-6 mRNA (qRT-PCR) from (A) after Remdesivir treatment (0.125–5 μM) of SARS-CoV-2-infected Calu-3 cells (MOI 0.04 TCID50/cell) with Remdesivir added 2 h prior to infection. Mean ± SEM, $n$ = 3.

E–H   Measurement of (E) viral genomic and subgenomic E RNA (F) fold induction of CXCL10, (G) IFIT2 and (H) IL-6 at 24 hpi, of Calu-3 cells with SARS-CoV-2 (MOI 0.04 TCID50$_{VERO}$/cell) with Remdesivir treatment (5 μM) prior to, at the time of, or 8 h post-infection. Mean ± SEM, $n$ = 3, one-way ANOVA with Dunnett's multiple comparisons test to compare to untreated infected condition ("mock"), ** ($P < 0.01$), *** ($P < 0.001$), **** ($P < 0.0001$).

I     Representative example of immunofluorescence staining of dsRNA (red) and N protein (green) after SARS-CoV-2 infection of Calu-3 at MOI 0.4 TCID50$_{VERO}$/cell, at time points shown. Nuclei (DAPI, blue). Scale bar represents 50 μm.

J     RNAi mediated depletion of MAVS, RIG-I or MDA5, reduced their expression levels as compared to siControl (siCtrl) Mean ± SEM, $n$ = 3. Two-way ANOVA with Sidak's multiple comparisons test, **** ($P < 0.0001$).

K–O   Fold induction of (K) IFNβ, (L) CXCL10, (M) IFIT2 (N) TNF and (O) IL-6 in SARS-CoV-2 infected Calu-3 cells (MOI 0.04 TCID50/cell) 24 hpi. Mean ± SEM, $n$ = 3, and compared to siCtrl for each gene by one-way ANOVA with Dunnett's multiple comparisons test, ** ($P < 0.01$), *** ($P < 0.001$), **** ($P < 0.0001$), n.s. : non-significant.

P     Live/dead stain counts for non-adherent cells, recovered at 48 hpi from supernatants of SARS-CoV-2 infected Calu-3 cells, depleted for MAVS or RNA sensors, compared to siCtrl. Non-adherent cell counts were determined by acquisition by flow cytometry for a defined period of time. Mean +/-SEM, $n$ = 3. Total numbers are compared with siCtrl by unpaired $t$-test, *** ($P < 0.001$).

Q–R   Viral E RNA and (R) released infectious virus (TCID50$_{VERO}$/cell) at 24 hpi of infected Calu-3 cells depleted for MAVs, RIG-I or MDA5. Mean ± SEM, $n$ = 3. Each group compared to siCtrl by one-way ANOVA with Dunnett's multiple comparisons test, *$P > 0.05$, ** ($P < 0.01$), n.s : non-significant.

MAVS abolished SARS-CoV-2-induced IL-6, CXCL10, IFNβ and IFIT2 gene expression (Fig 3J–O), consistent with RNA sensing being a key driver of SARS-CoV-2-induced innate immune activation. Concordantly, depletion of cytoplasmic RNA sensors RIG-I or MDA5 also reduced inflammatory gene expression after infection (Fig 3J–O). This suggested sensing of multiple RNA species given the different specificities of RIG-I and MDA5 (Hornung *et al*, 2006; Kato *et al*, 2006; Rehwinkel *et al*, 2010; Wu *et al*, 2013). Intriguingly, unlike RIG-I, MDA5 was not required for induction of NF-κB-sensitive genes IL-6 or TNF, consistent with differences in downstream consequences of RIG-I and MDA5 activation (Fig 3N and O) (Brisse & Ly, 2019). Abrogating SARS-CoV-2 sensing via MDA5 and MAVS depletion also reduced cell death, suggesting cell death is mediated by the host response rather than direct virus-induced damage (Fig 3P). Notably, sensor depletion did not strongly increase viral RNA levels (Fig 3Q), or the amount of released infectious virus (Fig 3R), confirming that innate immune activation via RNA sensing did not potently inhibit viral replication.

### NF-κB and JAK/STAT signalling drive innate immune responses

As a complementary approach to mapping SARS-CoV-2-induced innate immune activation, and to assess the potential of specific immunomodulators to impact inflammatory responses and viral replication, we examined the effect of inhibiting NF-κB activation using IK-β kinase (IKK-β) inhibitors TPCA-1 and PS1145. IKK-β is responsible for NF-κB p65 activation by phosphorylation following PRR signalling. Induction of ISGs and IL-6 was inhibited by TPCA-1, and with slightly reduced potency PS1145 (Figs 4A–C and EV5A and B). Inhibiting Janus kinase (JAK) with Ruxolitinib, to prevent JAK signalling downstream of the type I IFN receptor (IFNAR), also suppressed SARS-CoV-2-induced ISGs, but not NF-κB-sensitive IL-6 (Figs 4D–F and EV5C). Neither TPCA-1 nor Ruxolitinib treatment increased viral genome replication over a wide range of MOIs (Fig 4G and H) or N positivity or virion production after single dose infection (Fig EV5D–F). Importantly, NF-κB and JAK inhibition significantly reduced cell death in infected cultures (Fig 4I). This is consistent with our earlier observation and with the notion that the innate immune response to infection is the main driver of lung epithelial cell damage. Our data, thus far, show that SARS-CoV-2

infection of Calu-3 lung epithelial cells results in multi-pathway activation, driving pro-inflammatory and IFN-mediated innate immune responses that are inadequate or arise too late to restrict virus. Critically, they also suggest that SARS-CoV-2-induced IFN and pro-inflammatory gene expression can be therapeutically uncoupled from viral replication.

### Epithelial responses to SARS-CoV-2 drive macrophage activation

Resident and recruited pro-inflammatory macrophages in the lungs are associated with severe COVID-19 disease (Bost *et al*, 2020; Liao *et al*, 2020; Pairo-Castineira *et al*, 2020; preprint: Szabo *et al*, 2020). We therefore asked whether macrophages can support SARS-CoV-2 infection and how they respond indirectly to infection, through exposure to conditioned medium (CoM) from infected epithelial cells. Importantly, neither primary monocyte-derived macrophages (MDM) nor PMA-differentiated THP-1 cells (as an alternative macrophage model) supported SARS-CoV-2 replication, evidenced by lack of increase in viral RNA and by the absence of N-positive cells (Appendix Fig S1A–C). This is consistent with their lack of ACE2 and TMPRSS2 expression (Fig EV1A and B). However, exposure of MDM to virus-containing conditioned medium from infected Calu-3 cells (Fig 5A) led to significant macrophage ISG induction (Fig 5B, E and H) and increased expression of macrophage activation markers CD86 and HLA-DR (Fig 5C, D, F-G, I and J). Importantly, the immune stimulatory activity of conditioned media was dependent on RNA sensing and innate immune activation in infected Calu-3 cells because induction of inflammatory genes and macrophage activation markers was abolished by depletion of MAVS prior to Calu-3 infection (Fig 5B–D) or by inhibition of NF-κB (TPCA-1) or JAK activation (Ruxolitinib) in infected Calu-3 (Fig 5E–J). Note that in these experiments, MDM were exposed to equivalent numbers of viral genomes from the MAVS depleted, or inhibitor-treated conditioned media (Appendix Fig S1D–F). To confirm that soluble mediators produced by infected Calu-3 cells were key in driving MDM activation, we pre-treated MDM with either anti-IFN αβ receptor 2 (IFNAR2) antibody or Ruxolitinib to inhibit IFN signalling during exposure to CoM. Both treatments reduced induction of ISG IFIT2 and CXCL10 in MDM. We also saw a trend towards decreasing CCL5 expression but this was not significant, suggesting other pro-

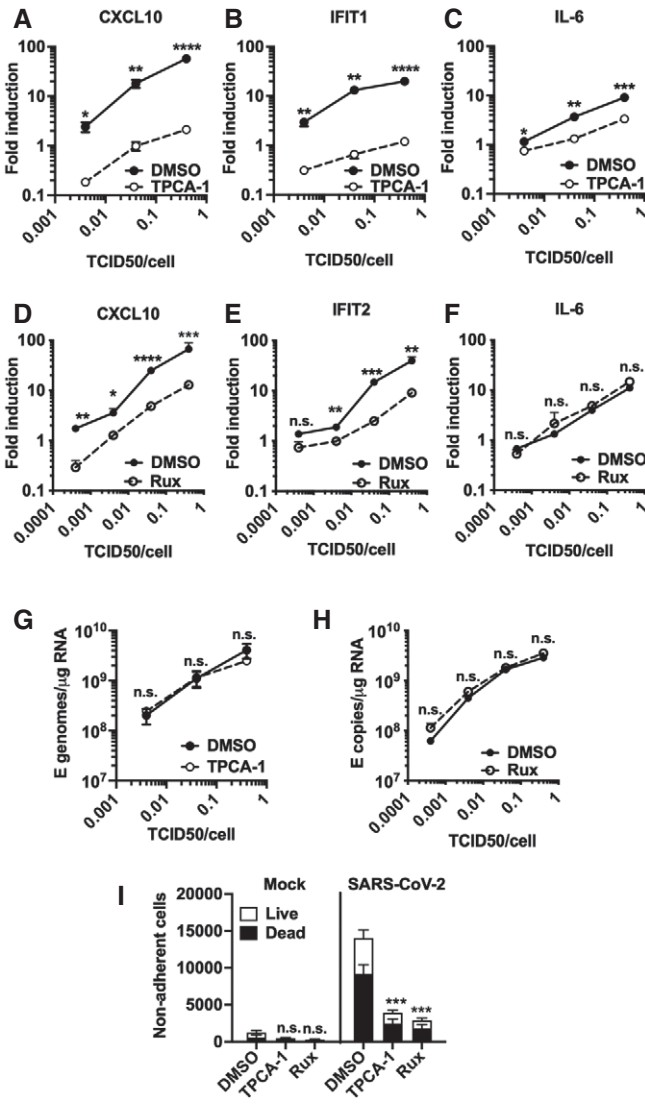

**Figure 4. NF-κB and JAK/STAT signalling drive innate immune responses.**

A–C Fold induction at 24 hpi of (A) CXCL10, (B) IFIT1 or (C) IL-6 mRNA (qRT-PCR) after Calu-3 infection with SARS-CoV-2 over a range of MOIs (0.004, 0.04, 0.4 TCID50$_{VERO}$/cell) in the presence of 10 μM TPCA-1 or DMSO as shown.

D–F Fold induction at 24 hpi of (D) CXCL10, (E) IFIT2 or (F) IL-6 mRNA (RT–qPCR) after Calu-3 infection with SARS-CoV-2 over a range of MOIs (0.0004, 0.004, 0.04, 0.4 TCID50$_{VERO}$/cell) in the presence of 2 μM Ruxolitinib (Rux) or DMSO as shown.

G, H Viral genomic and subgenomic E RNA at 24 hpi (RT–qPCR) with DMSO or TPCA (G) or Rux (H) treatment.

I Live/dead stain count from non-adherent cells recovered from supernatants of SARS-CoV-2-infected Calu-3 cultures (MOI 0.04 TCID50$_{VERO}$/cell) 48 hpi (flow cytometry). Mean ± SEM, (n = 3). One-way ANOVA comparison of inhibitor-treated infected cells to mock-treated infected cells. *** (P < 0.001).

Data information: (A-H) Mean ± SEM, n = 3, statistical comparisons are made by unpaired *t*-test comparing inhibitor-treated to mock-treated SARS-CoV-2-infected conditions at each MOI and each time point. * (P < 0.05), ** (P < 0.01), *** (P < 0.001), **** (P < 0.0001).

inflammatory mediators contributing to gene induction in MDM (Fig 5K–M, Appendix Fig S1G–I). Strikingly, inhibiting IFN signalling reduced the induction of MDM activation markers CD86 and HLA-DR underlining the importance of IFN in these responses to the infected Calu-3 supernatant (Fig 5N and O, Appendix Fig S1J and K). Together, these data demonstrate that production of IFNs and inflammatory mediators from infected lung epithelial cells, downstream of viral RNA sensing, can propagate potent pro-inflammatory macrophage activation.

## Pre-existing immune activation exacerbates SARS-CoV-2-dependent inflammation

Severe COVID-19 is associated with inflammatory co-morbidities, suggesting that pre-existing inflammatory states lead to inappropriate immune responses to SARS-CoV-2 and drive disease (Lucas *et al*, 2020; Mehra *et al*, 2020; Williamson *et al*, 2020; Wolff *et al*, 2020; Zhang *et al*, 2020). Macrophages in particular are thought to potentiate inflammatory responses in the lungs of severe COVID-19 patients (Nicholls *et al*, 2003; Liao *et al*, 2020) and so we investigated whether inflammatory stimuli might directly exacerbate macrophage responses to SARS-CoV-2 alone (Fig 6A–H). In these experiments, we produced virus in Caco-2, and therefore, it did not contain inflammatory cytokines (Fig EV1G). We detected low-level innate immune activation after exposure of MDM to SARS-CoV-2 alone (Fig 6B–H). However, when MDM were primed with 100 ng/ml LPS prior to exposure to SARS-CoV-2, we observed an enhanced response compared with exposure to virus or LPS alone, evidenced by significantly increased levels of ISGs (Fig 6D and E) and pro-inflammatory CCL5 (Fig 6C). Of note, LPS alone induced IL-6 and inflammasome-associated IL-1β expression and secretion and this was unaffected by virus exposure (Fig 6F–H). Exposure of MDM to SARS-CoV-2, prior to stimulation with LPS (Fig 6I–P), also enhanced macrophage inflammatory and ISG responses, but not IL-6 or IL-1β expression and secretion, compared to those detected with virus or LPS alone (Fig 6J–P). Importantly, LPS treatment of MDM, before or after virus challenge, did not alter SARS-CoV-2 permissivity of MDM, evidenced by no change in the level of detectable viral E gene in MDM supernatants (Fig 6B and J). Thus, the changes in MDM gene induction by virus after LPS treatment are due to differences in the MDM response to virus and not due to a difference in the amount of virus genome added or induction of viral gene expression.

Finally, we modelled the lung epithelial cell response to the cytokines observed in activated macrophages. We first selected IL-1β, as it was produced by LPS-treated, LPS-primed virus-exposed and virus primed LPS-exposed MDM (Fig 6G, H, O and P) and has been observed in severe COVID-19 patient lungs (Laing *et al*, 2020; Rodrigues *et al*, 2021). Treatment of Calu-3 with IL-1β during infection significantly increased induction of both ISGs and pro-inflammatory cytokines, compared to their induction by virus alone (Fig 6Q–T). The exception was IL-6, which was highly induced by virus even in the absence of IL-1β pre-treatment (Fig 6S). Next we treated Calu-3 cells with TNF, which is also produced by LPS-treated or primed MDM (Appendix Fig S2A and B) and implicated in

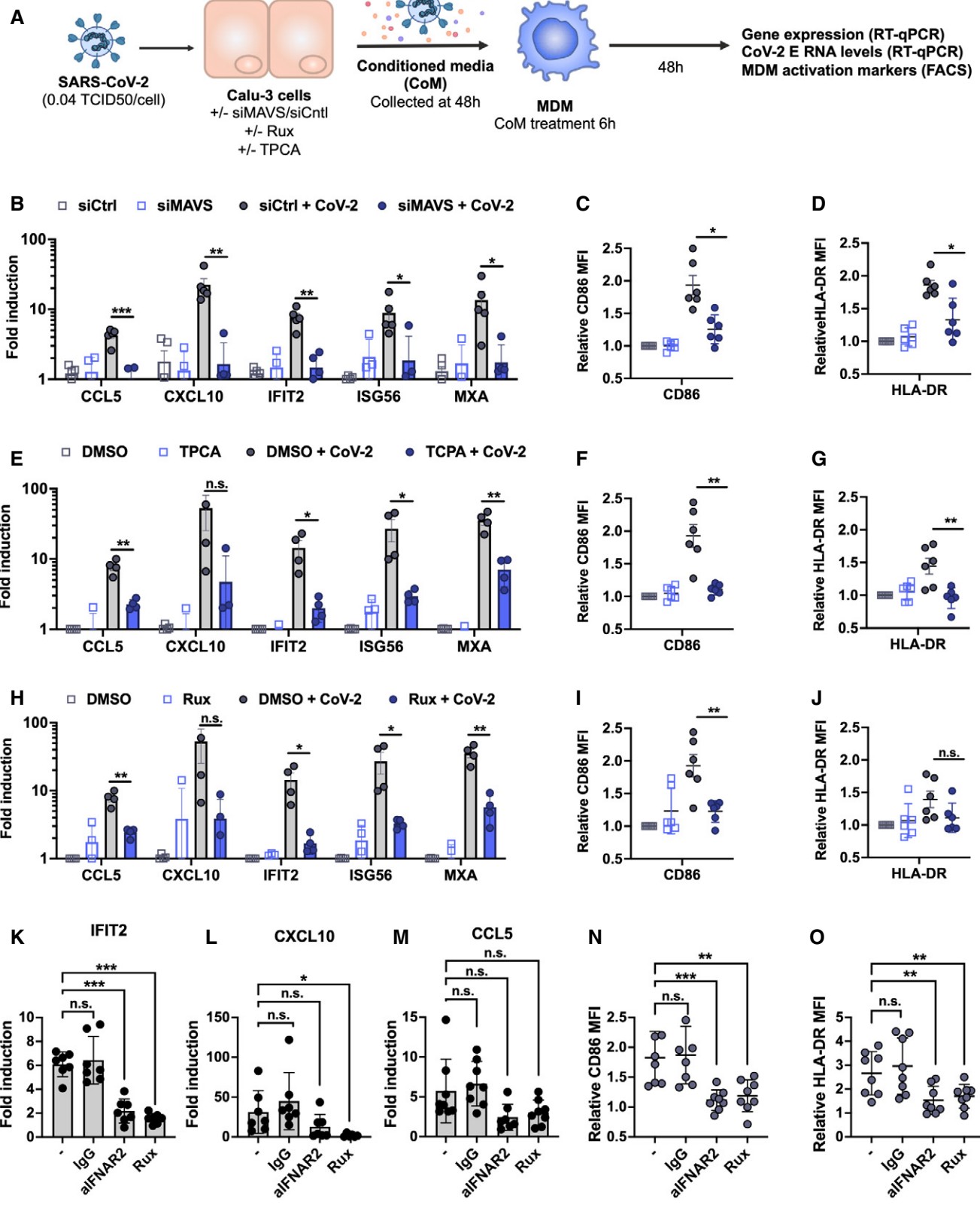

**Figure 5.**

◄

**Figure 5.  Epithelial responses to SARS-CoV-2 drive macrophage activation.**

A      Schematic of experimental design.

B–J   Calu-3 cells were transfected with siRNA targeting MAVS or non-targeting control (siCtrl) (B-D) or treated with DMSO vehicle or inhibitors 10 µM TPCA-1 (E-G) or 2 µM Ruxolitinib (Rux) (H-J) as shown, and were mock-infected or infected with SARS-CoV-2 at MOI 0.04 TCID50$_{VERO}$/cell. Virus-containing conditioned media (CoM) was harvested at 48 hpi. MDM were treated with Calu-3 virus-containing CoM for 6 hpi, before washing and measuring MDM gene expression (B, E, H), and MDM activation markers by flow cytometry 48 h later (C, D, F, G, I, J), plotting relative median fluorescent intensity (MFI) compared with mock-infected siCtrl (C, D) or mock-infected DMSO control (F, G, I, J). Legends in (B), (E) and (H) apply to (C, D), (F, G) and (I, J), respectively. The inhibitors in (E) and (H) were tested side-by-side with the same mock condition. Mean ± SEM shown, data from 4 to 6 independent MDM donors are shown. Statistical comparison by two-tailed paired *t*-test comparing MDM exposed to control infected CoM to siMAVS/inhibitor-treated infected CoM. * ($P < 0.05$), ** ($P < 0.01$), *** ($P < 0.001$).

K–O   MDM were treated with either anti-IFNAR antibody (2.5 µg/ml), an isotype control IgG antibody (IgG, 2.5 µg/ml), Rux (2 µM) or mock treated during 6 h of exposure to CoM from infected, unmodified Calu-3 cells, before washing and measuring MDM gene expression (K, L, M), and MDM activation markers (N, O) by flow cytometry 48 h later. Both gene expression and relative MFI are compared with mock-treated MDM exposed to CoM from uninfected Calu-3 cells. Mean ± SEM shown, data from 7 to 8 independent MDM donors are shown. Statistical testing by one-way paired ANOVA, comparing treated MDMs to untreated control by Dunnett's multiple comparison test, * ($P < 0.05$), ** ($P < 0.01$), *** ($P < 0.001$).

severe COVID-19 (Chua *et al*, 2020; Mahase, 2020), but found no enhancement of innate responses to SARS-CoV-2 (Appendix Fig S2C). However, both IL-1β and TNF treatment increased virus-induced epithelial cell death (Fig 6U and Appendix Fig S2D), without impacting viral replication (Fig 6V and Appendix Fig S2E). Together, these data suggest that SARS-CoV-2 infection of lung epithelium can promote immune activation of inflammatory macrophages, via secretion of cytokines, chemokines and virus from infected cells, and that this can be exacerbated by a pre-existing pro-inflammatory state. This is consistent with the hypothesis that chronic inflammatory states, rather than enhanced viral replication, drive detrimental immune activation and/or cell death.

## Discussion

We found that SARS-CoV-2 can replicate and spread effectively in lung epithelial Calu-3 cells over a wide range of inoculum doses despite inducing potent IFN responses and ISG expression. We propose that in the model system used here, innate immune activation occurs too late to suppress replication and attribute this to the virus deploying innate immune evasion and antagonism strategies early in infection. Indeed, coronaviruses replicate inside membranous vesicles, thought to protect viral RNA species from cytoplasmic sensing, and have complex capacity to antagonise innate immunity, including inhibition of MDA5 activation (Liu *et al*, 2021; Xia *et al*, 2020) and preventing nuclear entry of inflammatory transcription factors (Totura & Baric, 2012; Banerjee *et al*, 2020; Miorin *et al*, 2020; Park & Iwasaki, 2020; Yuen *et al*, 2020). Consistent with the literature, our data indicate that SARS-CoV-2 more effectively antagonises IRF3 activation and nuclear translocation than NF-κB. Indeed, it is possible that the innate immune response and the secreted signals produced by infected cells are dysregulated by viral manipulation and that this imbalanced response contributes to disease particularly in the context of underlying inflammatory pathology (Blanco-Melo *et al*, 2020; Giamarellos-Bourboulis *et al*, 2020; Lucas *et al*, 2020).

We demonstrate that SARS-CoV-2 can be sensed by both RIG-I and MDA5 and that, through their signalling adaptor MAVS, these sensors drive inflammatory responses in infected Calu-3 cells (Fig 7). Concordantly, both RIG-I and MDA5 have been implicated in sensing the murine coronavirus mouse hepatitis virus (Roth-Cross *et al*, 2008; Li *et al*, 2010) and MDA5 was recently shown to

sense SARS-CoV-2 and trigger IFN production (Rebendenne *et al*, 2021; Yin *et al*, 2021). Likewise, the activation of dsRNA sensor PKR has also been observed during SARS-CoV-2 infection of other cell types (Li *et al*, 2021). Interestingly, DNA sensing through cGAS-STING has also been reported to contribute to inflammatory responses (preprint: Neufeldt *et al*, 2020), likely through sensing of self-DNA or cellular damage. The eventual innate immune activation in Calu-3 cells is likely due to sensing of viral RNA when it accumulates to a level that overcomes sequestration and pathway inhibition by the virus, as well as to cellular stress responses to infection. Importantly, Calu-3 cells pre-treated with IFN resist infection illustrating that innate responses can suppress SARS-CoV-2 replication if an antiviral state is induced prior to infection, particularly with a low viral exposure dose.

Although SARS-CoV-2 RNA has been found associated with macrophages and monocytes from infected patients (Bost *et al*, 2020), we found that macrophages did not support SARS-CoV-2 replication. However, they were sensitive to conditioned media from infected Calu-3 containing virus, IFNs and pro-inflammatory mediators, inducing high levels of chemokine and ISG mRNA and expression of activation markers CD86 and HLA-DR upon exposure. Crucially, it is the response of the Calu-3 cells to virus infection, via RNA sensing, that drives macrophage activation in these experiments, evidenced by suppression of activation after either MAVS depletion or NF-κB (TPCA-1) or JAK inhibition (Ruxolitinib) in the infected Calu-3 cells. We found that IFNs produced by infected Calu-3 cells downstream of RNA sensing are key in driving MDM activation, evidenced by suppression of macrophage activation with IFNAR antibody, although we expect other soluble mediators to contribute. This inflammatory role for IFN may explain how delayed IFN response could contribute to pathogenicity rather than viral clearance (Park & Iwasaki, 2020).

A recent study suggested that sensing of abortive SARS-CoV-2 infection of macrophages may contribute to their activation (Zheng *et al*, 2021). Our data do not rule out a role for detection of abortive replication. However, they suggest that inflammatory mediators produced from infected cells, perhaps with responses particularly skewed towards pro-inflammatory pathways after viral manipulation, are key to macrophage activation. Notably, exposure of macrophages to infected Caco-2 supernatant, which contains virus but not significant levels of cytokine or IFN, did not strongly activate the macrophages. Indeed, our results show that it is important to distinguish between the effects of virus and the effects of cytokines in the

viral prep. Here, we have achieved this by using Caco-2 cells to produce virus without significant inflammatory cytokines and interferons and Calu-3 to produce virus with a corresponding inflammatory secretome.

Importantly, inhibiting RNA sensing or pathway activation did not particularly increase viral replication, consistent with our observation that, in this model at least, virus-induced innate immune responses do not significantly inhibit SARS-CoV-2 replication. These observations highlight the potential of immunomodulators in

reducing SARS-CoV-2-driven inflammatory disease. Indeed, suppression of JAK1/2 signalling with Baricitinib, in SARS-CoV-2-infected macaques, significantly reduced macrophage recruitment and inflammatory signatures and preliminary data support its use in COVID-19 (Bronte *et al*, 2020). These studies are consistent with epithelial-driven inflammation contributing to myeloid cell infiltration and the role of macrophages in exacerbating immune responses in COVID-19 (Giamarellos-Bourboulis *et al*, 2020; Hoang *et al*, 2020; Liao *et al*, 2020). Our data provide a framework for dissecting

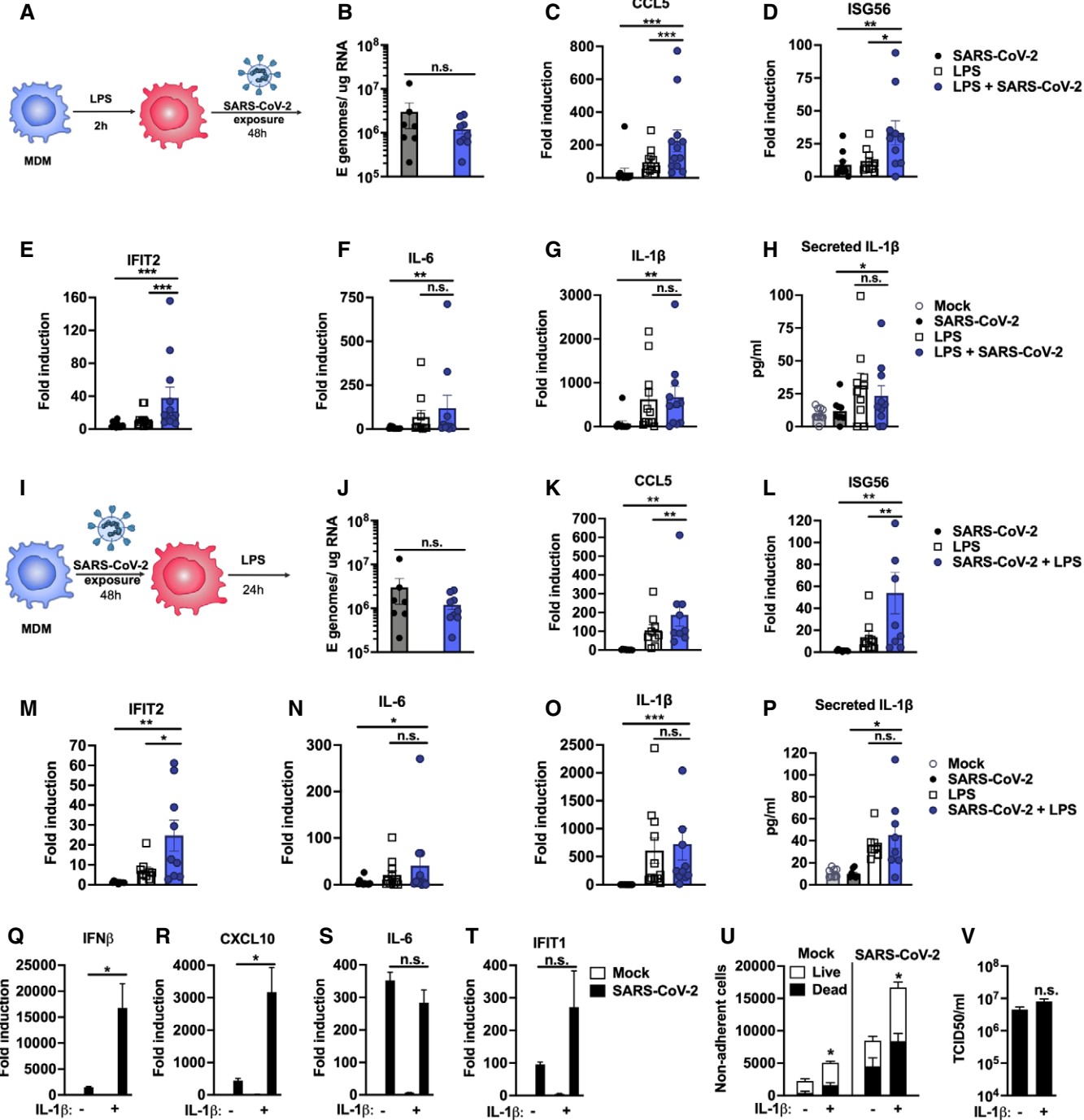

**Figure 6.**

**Figure 6.** **Pre-existing immune activation exacerbates SARS-CoV-2-dependent inflammation.**

A  Schematic of experimental design.
B–H  MDM were primed with 100 ng/ml LPS for 2 h before exposure to SARS-CoV-2 (MOI 0.02 TCID50$_{VERO}$/cell). (B) Expression of genomic and subgenomic viral E RNA at 48 h post-exposure in indicated conditions. (C-G) Host gene expression of (C) CCL5, (D) ISG56, (E) IFIT2, (F) IL-6 or (G) IL-1β mRNA was measured 48 hpi. (H) IL-1β secretion was detected in culture supernatants at 48 hpi by ELISA.
I  Schematic of experimental design. MDM were exposed to SARS-CoV-2 (MOI 0.02 TCID50$_{VERO}$/cell) for 48 h and subsequently stimulated with 100 ng/ml LPS for 24 h.
J–P  (J) Expression of genomic and subgenomic viral E RNA 72 h post-exposure in indicated conditions. (K–O) Host gene expression of (K) CCL5, (L) ISG56, (M) IFIT2, (N) IL-6 and (O) IL-1β mRNA. (P) IL-1β secretion was detected in culture supernatants at 48 hpi by ELISA.
Q–V  Calu-3 cells were infected with SARS-CoV-2 (MOI 0.04 TCID50$_{VERO}$/cell) in the presence or absence of 10 ng/ml IL-1β. (Q-T) Gene expression of (Q) IFNβ, (R) CXCL10, (S) IL-6 and (T) IFIT1 mRNA was measured after 24 h. Fold induction over untreated mock infection is shown, $n = 3$. (U) Non-adherent cells were collected at 48 h post-infection and live/dead stained. Cells were acquired by flow cytometry and cell counts determined by time-gating. For statistical comparison, total cell numbers were compared. (V) Viral release into culture supernatants at 24 h was measured by TCID50 on VeroE6 cells. (Q-V) Mock- and SARS-CoV-2-infected conditions were compared with or without IL1b-treatment, respectively, by unpaired *t*-test ($n = 3$). *$P < 0.05$; n.s., non-significant. Mean ± SEM shown.

Data Information: (A-P) Gene expression is shown as fold induction over untreated controls. Data from 8 to 13 independent donors are shown. Groups were compared as indicated by Wilcoxon matched-pairs signed-rank test, *$P < 0.05$, ** ($P < 0.01$), *** ($P < 0.001$).

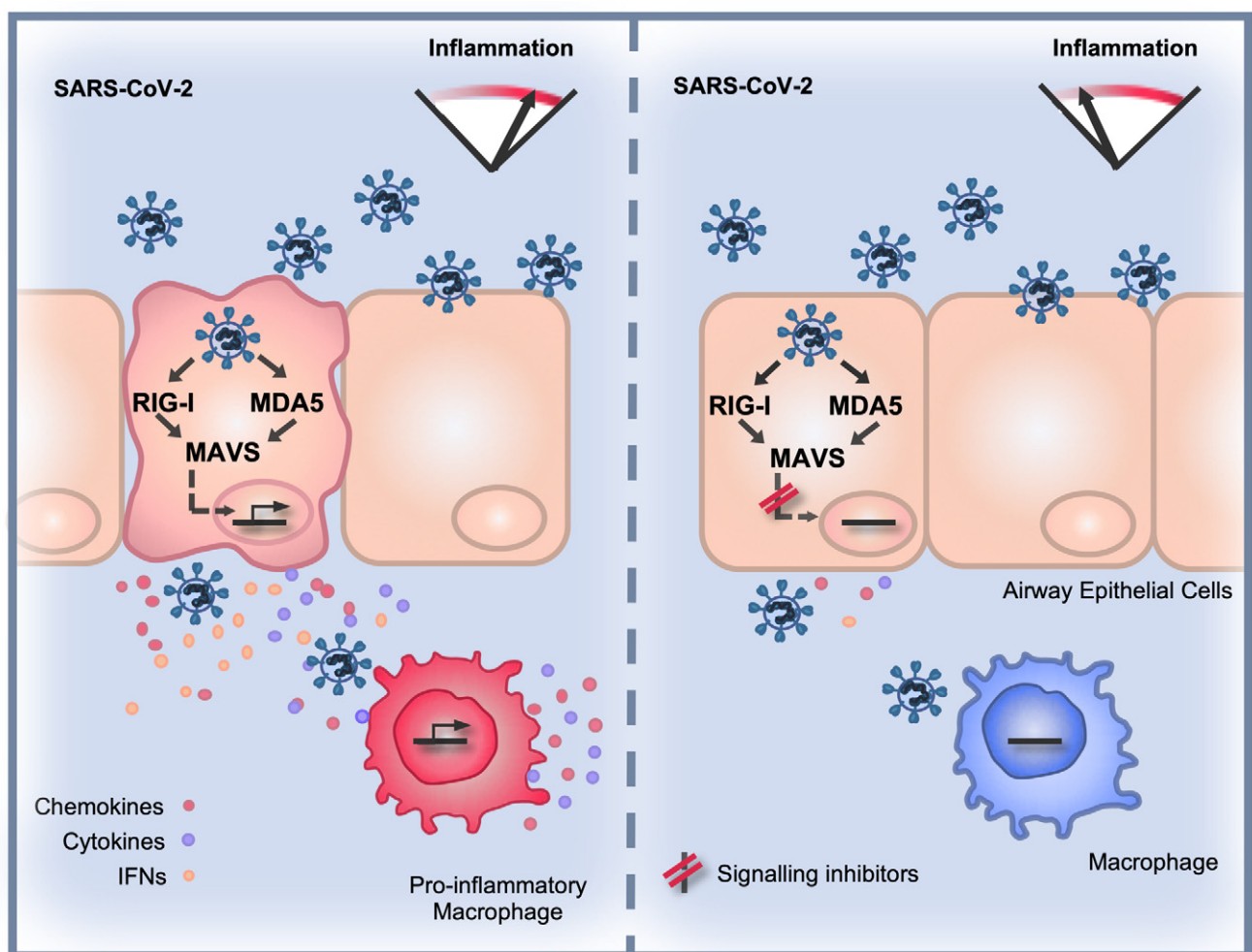

**Figure 7.** **SARS-CoV-2 induces a delayed inflammatory response that can be modified by specific pathway inhibitors.**

(Left) Infected lung epithelial cells sense SARS-CoV-2 RNA via cytoplasmic RNA sensors RIG-I and MDA5 to activate secretion of inflammatory mediators. Manipulation of RNA sensing early in infection by viral innate immune antagonists leads to a delayed and particularly inflammatory response. The infected cell secretome activates macrophages to potentiate a pro-inflammatory state at the site of infection. (Right) Inhibition of RNA sensing or downstream signalling pathways, for example with NF-κB inhibitors, reduces inflammation in infected cells and consequent activation of pro-inflammatory macrophages.

immunomodulators as therapeutics, and we propose that it is essential to test both immunomodulators, and direct-acting antivirals, in innate immune competent cells, rather than in Caco-2, Vero or other innate immune-inactive cell types, because the inevitable interactions between virus replication and innate immune pathways can influence drug efficacy and potency (Rasaiyaah *et al*, 2013; Kim *et al*, 2019; Sumner *et al*, 2020).

A key question is how our experiments in Calu-3 cells inform understanding of COVID-19. We propose that by studying virus replication in innate immune competent permissive host cells, we can probe the earliest interactions between the virus and the host that underpin subsequent inflammatory responses. Our data show that RNA sensing in infected Calu-3 cells creates a pro-inflammatory milieu capable of activating primary macrophages. Crucially, the combined profile of pro-inflammatory mediators in this system mirrors that observed in COVID-19 *in vivo* (Bost *et al*, 2020; Laing *et al*, 2020; preprint: Szabo *et al*, 2020) and primary airway epithelial cells (Fiege *et al*, 2021). We propose that *in vivo* it is the innate immune microenvironment in which the virus–host interaction occurs, and its consequent influence on immune activation, that determines disease outcome. This is consistent with our demonstration that exogenous inflammatory stimuli can drive a state in Calu-3 cells, and primary macrophages, that influences the response to virus, exacerbating inflammation. This link, between the immediate epithelial response to infection and external inflammatory signals, both amplified by macrophages, provides a plausible hypothesis to explain the association of severe COVID-19 with the presence of pro-inflammatory macrophages in bronchoalveolar lavage and patient lungs (Giamarellos-Bourboulis *et al*, 2020; Liao *et al*, 2020; preprint: Szabo *et al*, 2020) and inflammatory co-morbidities (Mehra *et al*, 2020; Williamson *et al*, 2020; Wolff *et al*, 2020), which could provide similar inflammatory stimulation.

It is remarkable how effective SARS-CoV-2 is in escaping human innate immune responses at the cellular level, despite being a recent zoonosis. Very low levels of adaptive change are consistent with adaptation to human replication prior to identification. Whether SARS-CoV-2 adapted in a non-human species prior to human infection, or whether adaptation in humans occurred before identification, remains unclear. One possibility is that coronaviruses replicate in a conserved niche, with regard to innate immune evasion, and thus are particularly good at zoonosis, perhaps evidenced by SARS-CoV-2 being preceded by SARS-CoV-1 and Middle Eastern Respiratory Syndrome virus (MERS), and apparent cross-species transfer and transmission in distantly related species including humans, bats (Boni *et al*, 2020), camels (Azhar *et al*, 2014), civet cats (Wang & Eaton, 2007) and mink (Koopmans, 2020).

Viral disease is often driven by host immune mechanisms that have evolved to protect the host from death, a paradox that is particularly evident in COVID-19. Here, we have taken a significant step towards explaining the consequences of SARS-CoV-2 infection of innate immune competent lung epithelial cells by illustrating how RNA sensing can directly drive potent inflammatory responses, irrespective of whether virus replication is suppressed. We propose that further studies addressing mechanisms of SARS-CoV-2 immune evasion and cytopathology, and the wider impact these have on epithelial-immune cell cross-talk, will inform development of effective therapeutics that are broadly active against zoonotic coronaviruses.

## Materials and Methods

### Cell culture and innate immune stimulation

Calu-3 cells (ATCC HTB-55) and Caco-2 cells were a kind gift from Dr Dalan Bailey (Pirbright Institute) and were originally obtained from ATCC. THP-1 dual cells were obtained from Invivogen. Vero.E6 was provided by NIBSC, Beas2B (ATCC CRL-9609) and Hulec5a (ATCC CRL-3244) were obtained from ATCC, and Detroit 562 (ATCC CCL-138) was a kind gift from Dr Caroline Weight (UCL). All cells tested negative for mycoplasma by commercial assay. All cells except THP-1 were cultured in Dulbecco's modified Eagle medium (DMEM) supplemented with 10% heat-inactivated FBS (Labtech), 100 U/ml penicillin/streptomycin, with the addition of 1% sodium pyruvate (Gibco) and 1% GlutaMax for Calu-3 and Caco-2 cells. All cells were passaged at 80% confluence. For infections, adherent cells were trypsinised, washed once in fresh medium and passed through a 70-μm cell strainer before seeding at $0.2 \times 10^6$ cells/ml into tissue-culture plates. Calu-3 cells were grown to 60–80% confluence prior to infection. THP-1 cells were cultured in RPMI (Gibco) supplemented with 10% heat-inactivated FBS (Labtech), 100 U/ml penicillin/streptomycin (Gibco), 25 mM HEPES (Sigma), 10 μg/ml of blasticidin (Invivogen) and 100 μg/ml of Zeocin™ (Invivogen). Caco-2 and Calu-3 cells were stimulated for 24 h with media containing TLR4 agonist Lipopolysaccharide (LPS) (PeproTech), the TLR3 agonist poly(I:C) (PeproTech) or the TLR7 agonist R837 (Invivogen), using the concentration stated on each figure. To stimulate RIG-I/MDA5 activation in Calu-3 cells, poly(I:C) was transfected. Transfection mixes were prepared using Lipofectamine 2000 (Invitrogen) in Opti-Mem (Thermo Fisher Scientific) according to the manufacturer's instructions.

### Isolation of primary monocyte-derived macrophages

Primary monocyte-derived macrophages (MDM) were prepared from fresh blood from healthy volunteers. The study was approved by the joint University College London/University College London Hospitals NHS Trust Human Research Ethics Committee, and written informed consent was obtained from all participants. Experiments conformed to the principals set out in WMA declaration of Helsinki and the Department of Health and Human Services Belmont Report. Peripheral blood mononuclear cells (PBMCs) were isolated by density gradient centrifugation using Lymphoprep (Stemcell Technologies). PBMCs were washed three times with PBS and plated to select for adherent cells. Non-adherent cells were washed away after 2 h and the remaining cells incubated in RPMI (Gibco) supplemented with 10% heat-inactivated pooled human serum (Sigma) and 100 ng/ml macrophage colony-stimulating factor (PeproTech). The medium was replaced after 3 days with RPMI with 5% FCS, removing any remaining non-adherent cells. Cells were infected or treated with conditioned media 3–4 days later.

### Virus culture and infection

SARS-CoV-2 strain BetaCoV/Australia/VIC01/2020 (NIBSC) was propagated by infecting Caco-2 cells at MOI 0.01 TCID50/cell, in DMEM supplemented with 2% FBS at 37°C. Virus was harvested at 72 h post-infection (hpi) and clarified by centrifugation at 2,100 *g*

for 15 min at 4°C to remove any cellular debris. Virus stocks were aliquoted and stored at −80°C. Virus titres were determined by 50% tissue-culture infectious dose (TCID50) on Vero.E6 cells. In brief, 96-well plates were seeded at $1 \times 10^4$ cells/well in 100 µl. Eight 10-fold serial dilutions of each virus stock or supernatant were prepared and 50 µl added to 4 replicate wells. Cytopathic effect (CPE) was scored at 5 days post-infection, and TCID50/ml was calculated using the Reed & Muench method (Reed & Muench, 1938), and an Excel spreadsheet created by Dr. Brett D. Lindenbach was used for calculating TCID50/ml values (Lindenbach, 2009).

For infections, multiplicities of infection (MOI) were calculated using TCID50/cell determining on Vero.E6 cells. Cells were inoculated with diluted virus stocks for 2 h at 37°C. Cells were subsequently washed twice with PBS, and fresh culture medium was added. At indicated time points, cells were harvested for analysis.

MDM were infected with virus diluted in RPMI, 5% FBS (estimated MOI 0.02 TCID50/cell). MDM were harvested at 24 h or 48 hpi for gene expression analysis. For priming experiments, MDM were stimulated with 100 ng/ml of LPS (HC4046, Hycult Biotech) for 2 h. Media was replaced, and cells were exposed to SARS-CoV-2 as before, diluted in RPMI, 5% FBS. Cells were collected after 48 h for analysis. Alternatively, cells were mock exposed or exposed to SARS-CoV-2 for 3 days and then stimulated with 100 ng/ml of LPS. Cells were harvested after 24 h for analysis.

In macrophage experiments, a minimum sample size of six independent experiments using cells derived from separate donors was used to give 90% power in order for a two-sided test to detect > twofold differences between two groups with an estimated standard deviation of 0.5.

## Sensor and adaptor depletion by RNAi

Calu-3 cells were transfected with 40 pmol of siRNA SMART pool against RIG-I (L-012511-00-0005), MDA5 (L-013041-00-0005), MAVS (L-024237-00-0005) or non-targeting control (D-001810-10-05) (Dharmacon) using Lipofectamine *RNAiMAX* Transfection Reagent (Invitrogen). Transfection medium was replaced after 24 h with DMEM medium supplemented with 10% FBS, 100 U/ml penicillin/streptomycin and cells cultured for additional 2 days. On day 3, cells were transfected again with the same siRNA smart pools. Transfection medium was replaced after 24 h and cells cultured for additional 2 days before infection. Gene depletion was verified using TaqMan Gene Expression Assay according to manufacturer's instructions detecting human RIG-I (FAM dye-labelled, TaqMan probe ID no. Hs01061436_m1), MAVS (FAM dye-labelled, TaqMan probe ID no. Hs00920075_m1), MDA5 (FAM dye-labelled, TaqMan probe ID no. Hs00223420_m1) or the housekeeping gene OAZ1 (FAM dye-labelled, TaqMan probe ID no. Hs00427923_m1).

## Treatment with cytokines, inhibitors and conditioned medium

Calu-3 cells were pre-treated with Remdesivir (Selleck Chemicals), TPCA-1 (Bio-Techne), PS1145 (Bio-Techne) or Ruxolitinib (Bio-Techne) at the indicated concentrations or DMSO control at an equivalent dilution for 1 h before SARS-CoV-2 infection unless otherwise stated. Inhibitors were maintained at the indicated concentrations throughout the experiments. For cytokine treatments, recombinant human IFNβ, IFNλ1, IFNλ2, IFNγ, IL1β or TNF

(PeproTech) at a final concentration of 10 ng/ml was added at the indicated time points. To generate conditioned media (CoM), Calu-3 cells were mock-infected or infected with SARS-CoV-2 at 0.04 TCID50/cell and supernatants were harvested 48 hpi, clarified by centrifugation at 2,100 *g* for 15 min and 4°C and stored at −80°C. For conditioned media experiments, MDM were exposed to CoM as indicated, which was diluted 1:5 in RPMI, 5% FBS. After 6 h, conditioned medium was replaced with RPMI, 5% FBS and cells were harvested at 48 h for gene expression and surface marker expression analysis. MDM were treated where indicated during CoM exposure with either 2 µM ruxolitinib (Bio-Techne) or 2.5 µg/ml anti-IFNAR antibody (pbl Assay Science) or an isotype control IgG2A antibody (R&D).

## RT–qPCR

RNA was extracted using RNeasy Micro Kits (Qiagen), and residual genomic DNA was removed from RNA samples by on-column DNAse I treatment (Qiagen). Both steps were performed according to the manufacturer's instructions. cDNA was synthesised using SuperScript III with random hexamer primers (Invitrogen). RT–qPCR was performed using Fast SYBR Green Master Mix (Thermo Fisher) for host gene expression or TaqMan Master mix (Thermo Fisher) for viral RNA quantification, and reactions performed on the QuantStudio 5 Real-Time PCR systems (Thermo Fisher). Host gene expression was determined using the 2-ΔΔCt method and normalised to GAPDH expression. Viral RNA copies were deduced by standard curve, using primers and a TaqMan probe specific for E, as described elsewhere (Corman *et al*, 2020) and below.

The following primers and probes were used as follows:

| Target | Sequence |
|---|---|
| ACE2 | Fwd 5′-CGAAGCCGAAGACCTGTTCTA-3′ <br> Rev 5′-GGGCAAGTGTGGACTGTTC-3′ |
| CCL5 | Fwd: 5′-CCCAGCAGTCGTCTTTGTCA-3′ <br> Rev 5′-TCCCGAACCCATTTCTTCTCT-3′ |
| CXCL10 | Fwd 5′-TGGCATTCAAGGAGTACCTC-3′ <br> Rev 5′-TTGTAGCAATGATCTCAACACG-3′ |
| GAPDH | Fwd 5′-GGGAAACTGTGGCGTGAT-3′ <br> Rev 5′-GGAGGAGTGGGTGTCGCTGTT-3′ |
| IFIT1/ISG56 | Fwd 5′-CCTCCTTGGGTTCGTCTACA-3′ <br> Rev 5′-GGCTGATATCTGGGTGCCTA-3′ |
| IFIT2 | Fwd 5′-CAGCTGAGAATTGCACTGCAA-3′ <br> Rev 5′-CGTAGGCTGCTCTCCAAGGA-3′ |
| IFNB1 | Fwd 5′-AGGACAGGATGAACTTTGAC-3′ <br> Rev 5′-TGATAGACATTAGCCAGGAG-3′ |
| IFNL1 | Fwd 5′-CACATTGGCAGGTTCAAATCTCT-3′ <br> Rev 5′-CCAGCGGACTCCTTTTTGG-3′ |
| IFNL3 | Fwd 5′-TAAGAGGGCCAAAGATGCCTT-3′ <br> Rev 5′-CTGGTCCAAGACATCCCCC-3′ |
| IL-1B | Fwd: 5′-CCTCCTTGGGTTCGTCTACA-3′ <br> Rev 5′-GGCTGATATCTGGGTGCCTA-3′ |
| IL-6 | Fwd 5′-AAATTCGGTACATCCTCGACG-3′ <br> Rev 5′-GGAAGGTTCAGGTTGTTTTCT-3′ |
| MX1 | Fwd 5′-ATCCTGGGATTTTGGGGCTT-3′ <br> Rev 5′-CCGCTTGTCGCTGGTGTCG-3′ |

Table (continued)

| Target | Sequence |
| --- | --- |
| *TMPRSS2* | Fwd 5′-CAAGTGCTCCAACTCTGGGAT-3′<br>Rev 5′-AACACACCGATTCTCGTCCTC-3′ |
| *TMPRSS4* | Fwd 5′-ATGCGGAACTCAAGTGGGC-3′<br>Rev 5′-CTGTTTGTCGTACTGGATGCT-3′ |
| *TNF* | Fwd 5′-AGCCTCTTCTCCTTCCTGATCGTG-3′<br>Rev 5′-GGCTGATTAGAGAGAGGTCCCTGG-3′ |
| SARS-CoV-2<br>E_Sarbeco_F | 5′-ACAGGTACGTTAATAGTTAATAGCGT-3′ |
| SARS-CoV-2<br>E_Sarbeco_Probe1 | 5′-FAM-ACACTAGCCATCCTTACTGCGCTTCG-TAMRA-3′ |
| SARS-CoV-2<br>E_Sarbeco_R | 5′-ATATTGCAGCAGTACGCACACA-3′ |

## Cytokine and LDH measurement

Secreted mediators were detected in cell culture supernatants by ELISA. CXCL10 and IL-6 protein were measured using DuoSet ELISA reagents (R&D Biosystems) according to the manufacturer's instructions.

Secreted lactate dehydrogenase (LDH) activity was measured as a correlate of cell death in culture supernatants using Cytotoxicity Detection Kit$^{PLUS}$ (Sigma) according to the manufacturer's instructions. Culture supernatants were collected at the indicated time points post-infection, clarified by centrifugation and stored at 4°C until LDH measurement.

## Antibodies

All antibody sources are cited with sample identifiers, and all antibodies were validated for their specific use by manufacturers or by previously published work as cited.

## Flow cytometry

For flow cytometry analysis, adherent cells were recovered by trypsinising or gentle scraping and washed in PBS with 2 mM EDTA (PBS/EDTA). Non-adherent cells were recovered from culture supernatants by centrifugation for 5 min at 300 *g* and washed once in PBS/EDTA. Cells were stained with fixable Zombie UV Live/Dead dye (BioLegend) for 6 min at room temperature. Excess stain was quenched with FBS-complemented DMEM. For MDMs, Fc-blocking was performed with PBS/EDTA+10% human serum for 10 min at 4°C. Cell surface with CD86-Bv711 (IT2.2, BioLegend) and HLA-DR-PerCpCy5.5 or PE-Cy7 (L243, BioLegend) staining was performed in PBS/EDTA at 4°C for 30min. Unbound antibody was washed off thoroughly, and cells were fixed in 4% PFA prior to intracellular staining. For intracellular detection of SARS-CoV-2 nucleoprotein, cells were permeabilised for 15 min with Intracellular Staining Perm Wash Buffer (BioLegend). Cells were then incubated with 1 µg/ml CR3009 SARS-CoV-2 cross-reactive antibody (a kind gift from Dr. Laura McCoy) in permeabilisation buffer for 30 min at room temperature, washed once and incubated with secondary Alexa Fluor 488-Donkey-anti-Human IgG (Jackson Labs). All samples were acquired on a BD Fortessa X20 or LSR II using BD FACSDiva software. Data were analysed using FlowJo v10 (Tree Star).

## Western blotting

For detection of ACE2 expression, whole cell protein lysates were separated by SDS–PAGE, transferred onto nitrocellulose and blocked in PBS with 0.05% Tween 20 and 5% skimmed milk. Membranes were probed with polyclonal goat anti-human ACE2 (1:500, AF933, R&D Biosystems) or rabbit anti-human beta-Actin (1:2,500, 6L12, Sigma) followed by donkey anti-goat IRdye 680CW or goat anti-rabbit IRdye 800CW (Abcam), respectively. Blots were imaged using an Odyssey Infrared Imager (LI-COR Biosciences) and analysed with Image Studio Lite software.

## Immunofluorescence microscopy and RNA-fluorescence *in situ* hybridisation

For imaging analysis, Calu-3 or Caco-2 cells were seeded and infected with SARS-CoV-2 in Optical 96-well plates (CellCarrier Ultra, PerkinElmer) and cells were fixed with 4% PFA at the indicated time points. Permeabilisation was carried out with 0.1% Triton X-100 (Sigma) in PBS for 15 min. A blocking step was carried out for 1 h at room temperature with 10% goat serum/1% BSA in PBS. Nucleocapsid (N) protein detection was performed by primary incubation with human anti-N antibody (Cr3009, 1 µg/ml) for 18 h, and washed thoroughly in PBS. Where appropriate, N protein staining was followed by incubation with rabbit anti- NF-κB (p65) (sc-372, Santa Cruz) or rabbit anti-IRF3 (sc-9082, Santa Cruz) for 1 h. Primary antibodies were detected by labelling with secondary anti-human AlexaFluor 488 and anti-rabbit AlexaFluor 546 conjugates (Jackson Immuno Research) for 1 h. For RNA-fluorescence *in situ* hybridisation (FISH), cells were immunofluorescently labelled for viral N protein (detected with AlexaFluor 488 or AlexaFluor 546 conjugates) followed by RNA visualisation using the ViewRNA Cell Plus Kit (Thermo Fisher). The ViewRNA probes implemented targeted *IL-6* (VA4-19075, AlexaFluor 488), *IFIT1* (VA4-18833, AlexaFluor 488) and *GAPDH* (VA1-10119, AlexaFluor 546). All cells were then labelled with HCS CellMask Deep Red (H32721, Thermo Fisher) and Hoechst33342 (H3570, Thermo Fisher). Images were acquired using the WiScan® Hermes High-Content Imaging System (IDEA Biomedical, Rehovot, Israel) at magnification 10×/0.4NA or 40×/0.75NA. Four-channel automated acquisition was carried out sequentially (DAPI/TRITC, GFP/Cy5). For 10× magnification, 100% density/100% well area was acquired, resulting in 64 FOV/well. For 40× magnification, 35% density/ 30% well area was acquired resulting in 102 FOV/well.

## Image analysis

NF-κB, IRF3, *IL-6* and *GAPDH* raw image channels were pre-processed using a batch rolling ball background correction in FIJI ImageJ software package (Schindelin *et al*, 2012) prior to quantification. Automated image analysis was carried out using the "Athena" HCS analysis software package (IDEA Biomedical IDEA Biomedical, Rehovot, Israel). For quantification of the percentage of nucleocapsid-positive cells within the population, the "Intracellular Granules" module was utilised. Nuclei were segmented using Hoechst33342 signal. Cell boundaries were determined by segmentation of CellMask signal. Infected cells were determined by thresholding intracellular N protein signal (Intracellular granules).

For all analysis, the N protein signal intensity was thresholded against the mock-infected wells to ensure no false segmentation of N +ve objects. Nuclear accumulation of NF-κB or IRF3 was carried out using the "Intranuclear Foci" module. Nuclei of cells were segmented using the Hoechst33342 stain. "Foci" of perinuclear N protein signal were identified and an "Infected" cell population determined based on the presence of such segmented foci objects. In all cells, the NF-κB or IRF3 signal present within segmented nuclei was quantified. For RNA-FISH quantification, the "Mitochondria" module was implemented. Nuclei were segmented using the Hoechst33342 stain. Cell cytoplasmic area was determined by segmentation of CellMask 647 signal. Intracellular N protein signal was segmented as "mitochondria" objects. *IL-6/GAPDH* RNA-FISH signal within segmented cells was then quantified. Infected cells were determined by the presence of N protein-segmented objects within the cell. Analysis parameters are detailed in Appendix Tables S1-S7.

### Statistical analysis

Statistical analysis was performed using GraphPad Prism. As indicated, normally distributed data were analysed for statistical significance by *t*-tests (when comparing two groups) or one-way ANOVA with Bonferroni or Dunnett's post-test (when comparing more than two groups). Wilcoxon ranked paired non-parametric tests were performed for primary macrophage data that were not normally distributed. For imaging analysis, where appropriate, integrated intensities were normalised to the mean intensity of the mock-infected population for that respective time point. Comparisons were made using a Kruskal–Wallis test with Dunn's multiple comparison. Data show the mean ± the SEM, where appropriate the median is shown, with significance shown on the figures, and levels were defined as $*P < 0.05$; $**P < 0.01$ and $***P < 0.001$, $****P < 0.0001$.

## Data availability

This study includes no data deposited in external repositories.

**Expanded View** for this article is available online.

## Acknowledgements
MN (207511)and CJ (108079) are funded by Wellcome Investigator Awards, and GJT was funded by a Wellcome Senior Fellowship (108183) followed by a Wellcome Investigator Award (220863). MVXW is supported by the NIHR Biomedical Research Centre at UCLH and IDEA BioMedical Ltd. Funds were also obtained from the University College London COVID-19 fund and the National Institutes of Health Research UCL/UCLH Biomedical Research Centre. We are grateful to Giada Mattiuzzo at NIBSC for SARS-CoV-2 and reagents, Laura McCoy at UCL for SARS-CoV-2N antibody, Dalan Bailey at The Pirbright Institute for cell lines and Richard Milne at UCL for valuable discussions and critical reading of the manuscript.

## Author contributions
LGT, A-KR, LZ-A, CJ and GJT conceptualised the study; LGT, A-KR, LZ-A, MVXW, JT, MN, CJ and GJT were involved in experimental set up, investigation and analysis; all authors wrote, reviewed and edited the article.

## Conflict of interest
The authors declare that they have no conflict of interest.

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
