## [Review Process File · The EMBO Journal]

SARS-CoV-2 sensing by RIG-I and MDA5 links epithelial infection to macrophage inflammation

Lucy Thorne, Ann Kathrin Reuschl, Lorena Zuliani-Alvarez, Matthew Whelan, Jane Turner, Mahdad Noursadeghi, Clare Jolly, and Greg Towers

DOI: [10.15252/embj.2021107826](https://doi.org/10.15252/embj.2021107826)

Corresponding author(s): Greg Towers (g.towers@ucl.ac.uk), Clare Jolly (c.jolly@ucl.ac.uk), Lucy Thorne (l.thorne@ucl.ac.uk)

Review Timeline:

Submission Date:	26th Jan 21
Editorial Decision:	23rd Feb 21
Revision Received:	30th Apr 21
Editorial Decision:	18th May 21
Revision Received:	24th May 21
Accepted:	28th May 21

Editor: Karin Dumstrei

Transaction Report:

Dear Greg,

Thank you for submitting your manuscript to The EMBO Journal. Your study has now been seen by the three referees and their comments are provided below.

As you can see from the comments below the response is a bit mixed. Referee #3 is not convinced that we get enough new insight for consideration here while referees #1 and 2 are more supportive. I do find that the study makes an important contribution and I would like to invite a revision.

Can we discuss how to best address the referees' comments as some of the comments are very well taken, but properly also beyond the scope of a revised version. We can do so via video or email whatever works best for you.

This is also a very competitive field with PMID 33440148 being published shortly before your manuscript was submitted to us and PMID 33514628 published during the review process. Please make sure to discuss both papers

When preparing your letter of response to the referees' comments, please bear in mind that this will form part of the Review Process File, and will therefore be available online to the community. For more details on our Transparent Editorial Process, please visit our website:

<https://www.embopress.org/page/journal/14602075/authorguide#transparentprocess>

Thank you for the opportunity to consider your work for publication. I look forward to discussing the revisions further with you

best Karin

Karin Dumstrei, PhD
Senior Editor
The EMBO Journal

- a point-by-point response to the referees' comments, with a detailed description of the changes made (as a word file).

- a word file of the manuscript text.
 - individual production quality figure files (one file per figure)
 - a complete author checklist, which you can download from our author guidelines (<https://www.embopress.org/page/journal/14602075/authorguide>).
 - Expanded View files (replacing Supplementary Information)
- Please see out instructions to authors
<https://www.embopress.org/page/journal/14602075/authorguide#expandedview>

The revision must be submitted online within 90 days; please click on the link below to submit the revision online before 24th May 2021.

Referee #1:

Thorne and colleagues analyzed sensing of SARS-CoV-2 infection by the interferon system and its consequences. They show that the virus replicates efficiently in Calu-3 cells and induces expression of IFNs, proinflammatory cytokines and ISGs. However, expression of IFN/ISGs is delayed as compared to viral replication and, in keeping with this finding, addition of type I IFN prior but to but not at the same time as virus or later reduces viral replication. Further, it is show that efficient induction of gene expression and cell killing but not viral replication depends on the activity of the viral RdRp and on RIG-I/MDA5. In addition, induction of ISG expression is shown to depend on an intact JAK/STAT pathway while IL-6 expression is blocked by inhibitors targeting NFkB. Finally, it is demonstrated that supernatants from infected Calu-3 cells promote macrophage activation and that this process is enhanced upon pre-activation of macrophages. The manuscript is well written and the results are of interest to the field, although some of them are not novel (PMID 33440148, 33514628). The following points should be addressed.

Major

While Calu-3 cells are a good model for infection of human lung epithelial cells, key findings - efficient IFN/ISG induction but no viral inhibition and proinflammatory phenotype of macrophages - should be confirmed with primary cells/organoids.

It is unclear whether the induction of the proinflammatory phenotype in macrophages was solely due to soluble mediators in the culture supernatants or whether abortive infection played a role. This should be investigated and PMID: 33277988 should be discussed.

Related to the point above, it would be important to know which factors in the Calu-3 cell supernatants were responsible for promoting the proinflammatory phenotype in macrophages. Can initial insights be obtained using antibodies directed against IFNs?

Minor

Can the authors exclude that IFN/ISGs have been induced in Caco-2 cells upon SARS-CoV-2 infection but with faster kinetics as compared to Calu-3? Maybe at 72 h post infection expression had already decreased to background levels.

Referee #2:

Thorne et al have explored the mechanism of innate sensing of SARS-CoV2 in epithelial cells, and also the potential link to pathological inflammation driven by immune cells in the infected lungs. The authors propose that the virus is sensed by RIG-like receptors in epithelial, to promote an innate immune response, which does not contribute to antiviral defense, but rather amplifies inflammation in macrophages. The results presented are based on well designed, performed, and presented data, and the conclusions are generally supported by the data. However, the physiological relevance of the data is difficult, because a relatively simple setup with epithelial cell lines and transfer of supernatants to macrophages is used. This part should be strengthened.

1. The proposed model that epithelial-cell-derived innate responses amplify pathological macrophage activities in the COVID19 lung should be tested. Systems that could be used include primary co-culture models, organoids, K18-hACE2 mice, existing deposited scRNA data.
2. The inhibitor data in Figure 4 should be complemented with test of the a TBK1 inhibitor.
3. The data in Fig 6 is somewhat disconnected from the rest of the paper, and does not "dig a little deeper" after the interesting data in Fig 5. The authors should do a serious effort to identify some of the factors in the virus-containing conditioned media that trigger macrophage activation.
4. Could also be interested to have the macrophage characterized with respect to whether the virus-containing conditioned media drive them in a M1, M2 or non-conventional activity direction.
5. In Fig. 2M, the authors show that IL6 responses are dependent on RIG-I and MAVS but not MDA5. To test whether the NF- κ B response is dependent on RIG-I exclusively, the authors should test for other NF κ B-induced genes, and also test for NF- κ B nuclear translocation (as in Fig 2A-B) in cells treated with the different RNAi.
6. For some of the presented data n=2. This should be improved to reach at least n=3 for all data presented.

Referee #3:

In this manuscript, the authors show replication of SARS-CoV-2 in human lung epithelial cells is

largely unaffected by the innate immune response. Through a careful timecourse analysis they provide evidence that the antiviral immune response is delayed until significant viral replication has already occurred, suggesting that there may be a threshold of replication that must occur for this virus to be robustly detected. They further confirm these findings with elegant single cell imaging analysis of NF- κ B, IRF3, and inflammatory mRNAs in infected and bystander cells. Additionally, they pinpoint the induction of type I IFN and inflammation to the RIG-I/MDA5 and MAVs signaling pathway. They then show that JAK/STAT and NF- κ B signaling inhibitors diminish inflammatory responses without major effects of SARS-CoV-2 replication. Lastly, they show that macrophages are not susceptible to infection by SARS-CoV-2, but can be activated by media from infected epithelial cells.

While the experiments are extensive, well controlled, and use relevant cells, the findings and concepts are not novel. It is unfortunate that so much high-quality effort has gone into rediscovering things that are already well established (e.g., epithelial cell infection by CoV-2 and resulting inflammation, RLR detection of CoV-2, involvement of NF- κ B and JAK/STATs in inflammatory and IFN responses to CoV-2, lack of macrophage infection by CoV-2, macrophage activation by cytokines). Given the lack of new insights provided by the work, I cannot recommend publication in a high impact journal.

Specific concerns/questions:

1. Knockdown efficiencies should be shown in the main text figures.
2. Figure 6, viral gene expression graphs are confusing given that it is stated that macrophages don't become infected.
3. The title of the paper/major conclusion is based on in vitro experiments with conditioned media or cytokines added to cultured macrophages. The same results with the limited panel of readouts would have been obtained with conditioned media from virtually any virus infection. SARS-CoV-2-specific insights or in vivo-relevant insights cannot be effectively gleaned from this work.

Referee #1:

Thorne and colleagues analyzed sensing of SARS-CoV-2 infection by the interferon system and its consequences. They show that the virus replicates efficiently in Calu-3 cells and induces expression of IFNs, proinflammatory cytokines and ISGs. However, expression of IFN/ISGs is delayed as compared to viral replication and, in keeping with this finding, addition of type I IFN prior but to but not at the same time as virus or later reduces viral replication. Further, it is show that efficient induction of gene expression and cell killing but not viral replication depends on the activity of the viral RdRp and on RIG-I/MDA5. In addition, induction of ISG expression is shown to depend on an intact JAK/STAT pathway while IL-6 expression is blocked by inhibitors targeting NFkB. Finally, it is demonstrated that supernatants from infected Calu-3 cells promote macrophage activation and that this process is enhanced upon pre-activation of macrophages. The manuscript is well written and the results are of interest to the field, although some of them are not novel (PMID 33440148, 33514628). The following points should be addressed.

Major

1. While Calu-3 cells are a good model for infection of human lung epithelial cells, key findings - efficient IFN/ISG induction but no viral inhibition and proinflammatory phenotype of macrophages - should be confirmed with primary cells/organoids.

This is a good suggestion for future work. Our key finding is that inflammation is directly downstream of RNA sensing, shown by suppressing pretty much all inflammatory responses measured by depleting RNA sensing machinery. We can't do this in primary airway cells because RNAi doesn't really work well. However, we already know that primary airway cells make inflammatory markers on CoV2 infection and we now cite this work in the following text in the discussion.

“Our data show that RNA sensing in infected Calu-3 cells creates a pro-inflammatory milieu capable of activating primary macrophages. Crucially the combined profile of pro-inflammatory mediators in this system mirrors that observed in COVID-19 *in vivo* (Bost et al., 2020;Laing et al., 2020;Szabo et al., 2020) and primary airway epithelial cells (Fiege et al., 2021)”.

2. It is unclear whether the induction of the proinflammatory phenotype in macrophages was solely due to soluble mediators in the culture supernatants or whether abortive infection played a role. This should be investigated and PMID: 33277988 should be discussed.

This is an excellent point and we have included the following text in the discussion and cited the paper,

“A recent study suggested that sensing of abortive SARS-CoV-2 infection of macrophages may contribute to their activation (Zheng et al., 2021). Our data do not rule out a role for detection of abortive replication. But they suggest that inflammatory mediators produced from infected cells, perhaps with responses particularly skewed towards pro-inflammatory pathways after viral manipulation, are key to macrophage activation. Notably, exposure of macrophages to infected Caco2 supernatant, which contains virus but not significant levels

of cytokine or IFN, did not strongly activate the macrophages. Indeed, our results show that it is important to distinguish between the effects of virus and the effects of cytokines in the viral prep. Here, we have achieved this by using Caco2 cells to produce virus without significant inflammatory cytokines and interferons and Calu3 to produce virus with a corresponding inflammatory secretome.”

3. Related to the point above, it would be important to know which factors in the Calu-3 cell supernatants were responsible for promoting the proinflammatory phenotype in macrophages. Can initial insights be obtained using antibodies directed against IFNs?

We agree that this is key. We have addressed this experimentally by treated the MDM with ruxolitinib and anti-IFNAR antibody to inhibit IFN signalling during exposure to the Calu-3 conditioned media. We have measured the same genes and macrophage activation markers, and the data suggest that IFN β produced by the Calu-3 cells contributes to MDM activation and the gene expression we have measured. We have included this data in Figure 5, given its importance in providing additional mechanistic insight, with controls in SupFig6, and modified the methods, results (pg 8-9) and discussion (pg12) text to reference it. This experiment has also addressed a similar point raised by reviewer 2 below.

Minor

4. Can the authors exclude that IFN/ISGs have been induced in Caco-2 cells upon SARS-CoV-2 infection but with faster kinetics as compared to Calu-3? Maybe at 72 h post infection expression had already decreased to background levels.

This is a really important point. In fact, we did this experiment right at the beginning of the study to make sure this wasn't a problem, but didn't include the data. We have now included these data showing no induction of gene expression at 24h post infection in Caco2s even at MOIs 500x higher than we use to prepare the viral stocks. These data are now included in supplementary figure 1 to clarify this point.

Referee #2:

Thorne et al have explored the mechanism of innate sensing of SARS-CoV2 in epithelial cells, and also the potential link to pathological inflammation driven by immune cells in the infected lungs. The authors propose that the virus is sensed by RIG-like receptors in epithelial, to promote an innate immune response, which does not contribute to antiviral defense, but rather amplifies inflammation in macrophages. The results presented a based on well designed, performed, and presented data, and the conclusions are generally supported by the data. However, the physiological relevance of the data is difficult, because a relatively simple setup with epithelial cell lines and transfer of supernatants to macrophages is used. This part should be strengthened.

1. The proposed model that epithelial-cell-derived innate responses amplify pathological

macrophage activities in the COVID19 lung should be tested. Systems that could be used include primary co-culture models, organoids, K18-hACE2 mice, existing deposited scRNA data.

Thank you for this suggestion, as explained above in answer to reviewer 1 we agree that in the long term these alternative systems will provide valuable insight to support our data. However, given primary airway experiments have been done by others, we are now citing these data in our discussion as follows

“Our data show that RNA sensing in infected Calu-3 cells creates a pro-inflammatory milieu capable of activating primary macrophages. Crucially the combined profile of pro-inflammatory mediators in this system mirrors that observed in COVID-19 *in vivo* (Bost et al., 2020; Laing et al., 2020; Szabo et al., 2020) and primary airway epithelial cells and primary airway epithelial cells (Fiege et al., 2021)”.

2. The inhibitor data in Figure 4 should be complemented with test of the a TBK1 inhibitor.
Thank you for this suggestion, we agree that there many inhibitors that we could have included such as a TBK1 inhibitor. As we have mapped the RNA sensors involved (Figure 3) and confirmed that inhibiting NFkB activation directly downstream of their activation as well as the second wave of IFN signalling dampens the innate and inflammatory response (Figure 4), we do not feel that including a TBK1 inhibitor will add significant mechanistic insight.

3. The data in Fig 6 is somewhat disconnected from the rest of the paper, and doe not "dig a later deeper" after the interesting data in Fig 5. The authors should do a serious effort to identify some of the factors in the virus-containing conditioned media that trigger macrophage activation.

This point is similar to that of reviewer one (point 3), and we have now addressed this point by demonstrating that IFN is a key driver in MDM activation. We inhibit IFN signalling in MDM with Ruxolitinib, or anti-IFNAR antibody, during exposure to infected Calu3 supernatant and show that activation markers and inflammatory gene expression are suppressed. We interpret these new data, presented in Fig 5k-O, as showing that IFN is a key feature of the proinflammatory response of infected cells. Failure to suppress all gene expression suggest further cytokines/chemokines also have a role.

4. Could also be interested to have the macrophage characterized with respect to whether the virus-containing conditioned media drive them in a M1, M2 or non-conventional activity direction.

This is a reasonable suggestion but we feel it is beyond the scope of the current manuscript.

5. In Fig. 2M, the autors show that IL6 responses are dependent on RIG-I and MAVS but not MDA5. To test whether the NF-kB response is dependent on RIG-I exclusively, the authors should test for other NFkB-induced genes, and also test for NF-kB nuclear translocation (as in Fig 2A-B) in cells treated with the different RNAi.

Thank you for this suggestion, we agree that this is interesting and we have measured TNF expression as another NFkB-dependent gene. We found the same result as for IL-6, that is was not affected by MDA5 depletion. These new data are included in figure 3 with the following text

“Concordantly, depletion of cytoplasmic RNA sensors RIG-I or MDA-5 also reduced inflammatory gene expression after infection (Figures 3J-N). This suggested sensing of multiple RNA-species given the different specificities of RIG-I and MDA5 (Hornung et al., 2006;Kato et al., 2006;Rehwinkel et al., 2010;Wu et al., 2013). Intriguingly, unlike RIG-I, MDA5 was not required for induction of NF- κ B-sensitive genes IL-6 or TNF, consistent with differences in downstream consequences of RIG-I and MDA5 activation (Figure 3N and O) (Brisse and Ly, 2019).”

6. For some of the presented data n=2. This should be improved to reach at least n=3 for all data presented.

This comment refers to the growth curves presented in figures 1 and 2. We have now performed this experiment again, clarifying in the figure legends that the data represent three independent experiments. We show two of the independent repeats between the main figure and supplementary figures. Please note that each growth curve derives from replicate wells and 3 or 4 MOIs to confirm our observations of replication timing versus innate response. This approach is more thorough than typical, as single MOI growth curve experiments are often presented and we are very confident that our findings are robust and representative. In the single cell analyses, we present data on thousands of cells for each repeat and condition, making it a very reliable dataset. Furthermore, the replication and innate response data is also repeated in experiments throughout the paper, the timing for which were based on the growth curve data, which are all consistent and confirm the reliability of the phenotypes described.

Referee #3:

In this manuscript, the authors show replication of SARS-CoV-2 in human lung epithelial cells is largely unaffected by the innate immune response. Through a careful timecourse analysis they provide evidence that the antiviral immune response is delayed until significant viral replication has already occurred, suggesting that there may be a threshold of replication that must occur for this virus to be robustly detected. They further confirm these findings with elegant single cell imaging analysis of NF- κ B, IRF3, and inflammatory mRNAs in infected and bystander cells. Additionally, they pinpoint the induction of type I IFN and inflammation to the RIG-I/MDA5 and MAVs signaling pathway. They then show that JAK/STAT and NF- κ B signaling inhibitors diminish inflammatory responses without major effects of SARS-CoV-2 replication. Lastly, they show that macrophages are not susceptible to infection by SARS-CoV-2, but can be activated by media from infected epithelial cells.

While the experiments are extensive, well controlled, and use relevant cells, the findings and

concepts are not novel. It is unfortunate that so much high-quality effort has gone into rediscovering things that are already well established (e.g., epithelial cell infection by CoV-2 and resulting inflammation, RLR detection of CoV-2, involvement of NF-kB and JAK/STATs in inflammatory and IFN responses to CoV-2, lack of macrophage infection by CoV-2, macrophage activation by cytokines). Given the lack of new insights provided by the work, I cannot recommend publication in a high impact journal.

Specific concerns/questions:

1. Knockdown efficiencies should be shown in the main text figures.

The knockdown efficiencies are already shown in the main figure 3.

2. Figure 6, viral gene expression graphs are confusion given that it is stated that macrophages don't become infected.

We are sorry if this was not clear. Viral gene PCR is performed to show that we see the same amount of viral genome, ie dose of virus, when we're comparing host gene expression in LPS treated and untreated target cells. This is important, because it allows us to be sure the difference in response is due to a difference in the behaviour of the target cells rather than a different dose of virus being used, or induction of viral replication by LPS. We have clarified this point with the following text,

“Importantly, LPS treatment of MDM, before or after virus challenge, did not alter SARS-CoV-2 permissivity of MDM, evidenced by no change in the level of detectable viral E gene in MDM supernatants (Figure 6B and J). Thus, the changes in MDM gene induction by virus after LPS treatment are due to differences in the MDM response to virus and not due to a difference in the amount of viral genome added or induction of viral gene expression.”

3. The title of the paper/major conclusion is based on in vitro experiments with conditioned media or cytokines added to cultured macrophages. The same results with the limited panel of readouts would have been obtained with conditioned media from virtually any virus infection. SARS-CoV-2-specific insights or in vivo-relevant insights cannot be effectively gleaned from this work.

We do not think that any viral infection would lead to the same cocktail of soluble mediators produced in epithelial cells. This is certainly not true for the other viruses we work on, eg HIV and HCV. We make the case in the discussion that this response is a result of the unique combination of evasion and antagonism strategies employed by SARS-CoV-2 to counter the innate response and permit human to human transmission. A particular manipulation of IRF3, leaving NF-kB pathways intact to drive inflammation, is reflected in our transcription factor translocation data in Fig 2. We observed significantly higher activation of NFkB than IRF3, suggesting SARS-CoV-2 more effectively inhibits IRF3 activation, consistent with the current literature. To clarify these points we have added the following text to the results and discussion sections.

Results

“IRF3 activation was also virus dose dependent but did not maximise until 72 hpi, later than NF- κ B (Figures 2C and D, Figure S3), and we observed a more modest shift in IRF3 nuclear intensity compared to NF- κ B throughout infection. These data are consistent with the requirement of a threshold of viral RNA replication to induce transcription factor translocation and innate immune activation, and suggest that SARS-CoV-2 may antagonise IRF3 activation to a greater extent than NF- κ B.”

Discussion

“Consistent with the literature, our data indicate that SARS-CoV-2 more effectively antagonises IRF3 activation and nuclear translocation than NF- κ B. Indeed, it is possible that the innate immune response and the secreted signals produced by infected cells are dysregulated by viral manipulation, and that this imbalanced response contributes to disease particularly in the context of underlying inflammatory pathology (Blanco-Melo et al., 2020; Giamarellos-Bourboulis et al., 2020; Lucas et al., 2020).”

Dear Greg,

Thanks for submitting your manuscript to The EMBO Journal.

Your study has now been seen by referee #1 and as you can see from the comments below, the referee appreciates the introduced changes and support publication here.

I am therefore very happy to let you know that we will accept the manuscript for publication here. There are just a few remaining editorial issues to resolve before I can send you the formal accept letter

- we are missing 3-5 keywords
- we need a Data Availability section. This is the place to enter accession numbers etc. As far as I can see no data is generated that needs to be deposited in a database. If this is correct please state: This study includes no data deposited in external repositories. Please place it after the Materials and methods and before Acknowledgements
- We need a conflict of interest statement
- please double check the reference format. Author list should be capped after 10 authors
- we are missing an ORCID ID for Lucy - please see guide to authors
- I think that the grant numbers are missing from the manuscript text
- Callouts to Fig 2G and 3R are missing. There is a callout to supplemental table 1, but no table
- "Methods" needs correcting to Materials and Methods
- Fig EV3 only has one panel so does not need the 'A' label.
- "Extended view Figure Legends" should be corrected to Expanded view Figure Legends.
- Our publisher has also done their pre-publication check on your manuscript. When you log into the manuscript submission system you will see the file "Data edited manuscript file". Please take a look at the word file and the comments regarding the figure legends and respond to the issues.
- We include a synopsis of the paper (see <http://emboj.embopress.org/>). Please provide me with a general summary statement and 3-5 bullet points that capture the key findings of the paper.
- We also need a summary figure for the synopsis. The size should be 550 wide by [200-400] high (pixels). You can also use something from the figures if that is easier.

That should be all - let me know if you have any further questions

Best Karin

Karin Dumstrei, PhD
Senior Editor
The EMBO Journal

Further information is available in our Guide For Authors:

The revision must be submitted online within 90 days; please click on the link below to submit the revision online before 16th Aug 2021.

Referee #1:

The authors have adequately addressed the points raised by this reviewer. The demonstration that antibodies targeting components of the IFN system blocked expression of macrophage activation markers provided additional mechanistic insights and strengthened the study.

Please find below a point-by-point response addressing all of the changes required for publication.

- we are missing 3-5 keywords

Keywords have been added to the start of the manuscript:

'SARS-CoV-2, inflammation, RNA sensing, epithelial, macrophage'

- we need a Data Availability section. This is the place to enter accession numbers etc. As far as I can see no data is generated that needs to be deposited in a database. If this is correct please state: This study includes no data deposited in external repositories. Please place it after the Materials and methods and before Acknowledgements

This section and sentence has been added to the manuscript.

- We need a conflict of interest statement

The following text has been added to the manuscript after the materials and methods section:

'Conflict of Interest Statement

The authors have no conflicts of interest to declare.'

- please double check the reference format. Author list should be capped after 10 authors

The format has been updated to EMBO J and author lists are now capped at 10 authors.

- we are missing an ORCID ID for Lucy - please see guide to authors

I have added this, apologies I thought I had already!

- I think that the grant numbers are missing from the manuscript text

The grant numbers have been added to the manuscript:

- Callouts to Fig 2G and 3R are missing. There is a callout to supplemental table 1, but no table

We have modified the text to reference Fig2G: *'IFIT1 transcripts (a prototypic ISG) measured by FISH also demonstrated rapid induction in N-positive cells with increased signal from at 6 hpi (Figure 2G and H).'*

We have modified the text to reference Fig3R and correct the figure 3 call outs: *'Depletion of RNA sensing adaptor MAVS abolished SARS-CoV-2-induced IL-6, CXCL10, IFN β and IFIT2 gene expression (Figures 3 J-O), consistent with RNA sensing being a key driver of SARS-CoV-2-induced innate immune activation. Concordantly, depletion of cytoplasmic RNA sensors RIG-I or MDA-5 also reduced inflammatory gene expression after infection (Figures 3J-O). This suggested sensing of multiple RNA-species given the different specificities of RIG-I and MDA5 (Hornung et al, 2006; Kato et al, 2006; Rehwinkel et al, 2010; Wu et al, 2013). Intriguingly, unlike RIG-I, MDA5 was not required for induction of NF- κ B-sensitive genes IL-6 or TNF, consistent with differences in downstream consequences of RIG-I and MDA5 activation (Figure 3N and O) (Brisse & Ly, 2019). Abrogating SARS-CoV-2 sensing via MDA5 and MAVS depletion also reduced cell death, suggesting cell death is mediated by the host*

response rather than direct virus-induced damage (Figure 3P). Notably, sensor depletion did not strongly increase viral RNA levels (Figure 3Q), or the amount of released infectious virus (Figure 3R), confirming that innate immune activation via RNA sensing did not potently inhibit viral replication.'

We have now provided file for Supplementary Tables 1A-G.

- **"Methods" needs correcting to Materials and Methods**

This has been changed in the manuscript to 'Materials and Methods'.

- **Fig EV3 only has one panel so does not need the 'A' label.**

The 'A' label has been removed and a modified figure provided.

- **"Extended view Figure Legends" should be corrected to Expanded view Figure Legends.**

This has been changed in the manuscript to Expanded view Figure Legends.

- **Our publisher has also done their pre-publication check on your manuscript. When you log into the manuscript submission system you will see the file "Data edited manuscript file". Please take a look at the word file and the comments regarding the figure legends and respond to the issues.**

We have checked and corrected each of the comments in the data edited manuscript file which all related to the figure legends. We have modified these in the updated, newly uploaded manuscript.

- **We include a synopsis of the paper (see <http://emboj.embopress.org/>). Please provide me with a general summary statement and 3-5 bullet points that capture the key findings of the paper.**

We have included the summary at the start of the manuscript:

'SARS-CoV-2 induces a robust, delayed innate immune response in airway epithelial cells, driven by activation of RNA sensors, which propagates inflammation through macrophage activation.'

The following bullet points have been added to the start of the manuscript and are provided below:

'Key points:

- *SARS-CoV-2 activates RNA sensors and consequent inflammatory responses in lung epithelial cells.*
- *Epithelial RNA sensing responses drive pro-inflammatory macrophage activation.*
- *Exogenous inflammatory stimuli exacerbate responses to SARS-CoV-2 in both epithelial cells and macrophages.*
- *Immunomodulators inhibit RNA sensing responses and consequent macrophage inflammation.'*

- **We also need a summary figure for the synopsis. The size should be 550 wide by [200-400] high (pixels). You can also use something from the figures if that is easier.**

We would like to use Figure7 as the summary figure for the synopsis if this is possible?

the grey bars in Fig 5E and H look identical - did you use the same sample or is it a mistake. If you used the same samples please juts make sure that this is clearly indicated in the figure legend.

Yes we performed the inhibitors in FigE (Rux) and Fig5H (TPCA1) side-by-side so the mocks are the same, we have made this clear in the figure legend:

'The inhibitors in (E) and (H) were tested side-by-side with the same mock condition.'

The Mock lane in Fig EV3 is the same within panels for NF-Kbeta and within the IRF3 panels. I presume that is how the experiment was done, but please state this clearly in the figure legend.

We have modified the figure legend to state that the conditions were all performed side-by-side with the same mock:

'All MOIs and mock were performed side-by-side and the mock is the same within panels for NF-Kbeta and within the IRF3 panels.'

Dear Greg,

Thank you for submitting your manuscript to The EMBO Journal. I have now had a chance to take a careful look at everything and all looks good.

I am therefore very pleased to accept the manuscript for publication here.

Congratulations on a nice study!

With best wishes

Karin

Karin Dumstrei, PhD
Senior Editor
The EMBO Journal

Please note that it is EMBO Journal policy for the transcript of the editorial process (containing referee reports and your response letter) to be published as an online supplement to each paper. If you do NOT want this, you will need to inform the Editorial Office via email immediately. More information is available here: https://emboj.embopress.org/about#Transparent_Process

Your manuscript will be processed for publication in the journal by EMBO Press. Manuscripts in the PDF and electronic editions of The EMBO Journal will be copy edited, and you will be provided with page proofs prior to publication. Please note that supplementary information is not included in the proofs.

Should you be planning a Press Release on your article, please get in contact with embojournal@wiley.com as early as possible, in order to coordinate publication and release dates.

If you have any questions, please do not hesitate to call or email the Editorial Office. Thank you for your contribution to The EMBO Journal.

Corresponding Author Name: Greg J Towers

Manuscript Number: EMBOJ-2021-107826